# Noise-Adaptive Thompson Sampling for Linear Contextual Bandits

**Ruitu Xu**
Department of Statistics and Data Science
Yale University
New Haven, CT 06511
`ruitu.xu@yale.edu`

**Yifei Min**
Department of Statistics and Data Science
Yale University
New Haven, CT 06511
`yifei.min@yale.edu`

**Tianhao Wang**
Department of Statistics and Data Science
Yale University
New Haven, CT 06511
`tianhao.wang@yale.edu`

## Abstract

Linear contextual bandits represent a fundamental class of models with numerous real-world applications, and it is critical to developing algorithms that can effectively manage noise with unknown variance, ensuring provable guarantees for both worst-case constant-variance noise and deterministic reward scenarios. In this paper, we study linear contextual bandits with heteroscedastic noise and propose the first noise-adaptive Thompson sampling-style algorithm that achieves a variance-dependent regret upper bound of $\widetilde{\mathcal{O}}\left(d^{3/2} + d^{3/2}\sqrt{\sum_{t=1}^{T}\sigma_t^2}\right)$, where $d$ is the dimension of the context vectors and $\sigma_t^2$ is the variance of the reward in round $t$. This recovers the existing $\widetilde{\mathcal{O}}(d^{3/2}\sqrt{T})$ regret guarantee in the constant-variance regime and further improves to $\widetilde{\mathcal{O}}(d^{3/2})$ in the deterministic regime, thus achieving a smooth interpolation in between. Our approach utilizes a stratified sampling procedure to overcome the too-conservative optimism in the linear Thompson sampling algorithm for linear contextual bandits.

## 1 Introduction

Linear contextual bandits represent a natural extension of multi-armed bandit problems (Auer et al., 2002a; Robbins, 1952; Lai et al., 1985), where the reward of each arm is assumed to be a linear function of the contextual information associated with the arm. Such problems manifest in numerous real-world applications, encompassing online advertising (Wu et al., 2016), recommendation systems (Deshpande and Montanari, 2012), and personalized medicine (Varatharajah and Berry, 2022; Lu et al., 2021). A multitude of algorithms have been proposed, tailored to diverse settings within the domain of linear contextual bandits (Auer et al., 2002b; Abe et al., 2003). Notably, two main streams of approaches to address the exploration-exploitation dilemma in linear contextual bandits have emerged: Upper Confidence Bound (UCB) (Chu et al., 2011; Abbasi-Yadkori et al., 2011) and Thompson sampling (TS) (Agrawal and Goyal, 2013; Abeille and Lazaric, 2017). In practical scenarios, the varying and non-transparent noise variance inherent in each reward is a common phenomenon (Towse et al., 2015; Omari et al., 2018; Somu et al., 2018; Cheng and Kleijnen, 1999). Such noise, which may exhibit heteroscedasticity and correlation with the context, can significantly impact algorithm performance, particularly when the variance of the noise is unknown a priori (Kirschner

37th Conference on Neural Information Processing Systems (NeurIPS 2023).

and Krause, 2018; Zhao et al., 2022). Thus, it is imperative to develop adaptive algorithms capable of handling noise with unknown heteroscedastic variance, ensuring provable guarantees when operating with both constant-variance noise and deterministic reward scenarios. Recently, a few variance-aware UCB algorithms have been proposed for linear bandits. In particular, Zhou et al. (2021) examined the scenario where the variance is known and observed after each arm pull, while Zhang et al. (2021); Kim et al. (2022b); Zhao et al. (2023) explored the unknown variance case. Nevertheless, there is a scarcity of results concerning TS algorithms in this context.

Thompson Sampling (Thompson, 1933), a notable Bayesian approach, provides a computationally efficient means of addressing the exploration-exploitation trade-off (Rusmevichientong and Tsitsiklis, 2010; Chapelle and Li, 2011). This method applies posterior sampling to generate a parameter estimator for arm selection, thereby naturally balancing exploration and exploitation by selecting arms with high expected rewards and those with substantial uncertainty (Russo et al., 2018; Riquelme et al., 2018). In the existing literature, the regret upper bound for the linear TS algorithm for linear contextual bandits is of order $\widetilde{\mathbb{O}}(d^{3/2}\sqrt{T})$ (Agrawal and Goyal, 2013; Abeille and Lazaric, 2017), and the existing algorithm cannot adapt to higher-order structure of the noise.

As an initial endeavor to incorporate variance information into TS-style algorithmic design for linear bandits, we adopt a weighted ridge regression approach to construct a TS algorithm, `LinVDTS`, detailed in Algorithm 2. This algorithm achieves a variance-dependent performance guarantee on the expected regret, as demonstrated in Theorem 4.1. Nonetheless, a technical barrier intrinsic to TS-style algorithms, yet absent in UCB, emerges during this naive integration of weighted regression with TS. Specifically, the issue stems from inadequate control over the variance-adjusted norm of unselected context vectors at each step, necessitating an anti-concentration argument that only allows for a variance-dependent upper bound on the expected regret and precludes the acquisition of a high-probability guarantee that is simultaneously variance-dependent. More discussion on this will be given in Section 4.2.

In a bid to navigate this intricate issue and exert explicit uncertainty control, we further propose a novel noise-adaptive TS variant in Algorithm 3. Our algorithmic design features (1) an uncertainty stratification scheme applied to contextual vectors, providing explicit regulation of the uncertainty quantification, (2) a cascading construction process for the feasible set, which, in tandem with the stratification scheme, operates to filter out unlikely arms, thereby eliminating the requirement for the anti-concentration argument in the analysis, and (3) the use of noise-adaptive confidence radii to direct the sampling procedure, thereby ensuring a more constricted variance-dependent regret. The theoretical analysis yields a high-probability regret bound that interpolates smoothly between the constant-variance and deterministic reward regimes.

**Main contributions.**    The main contributions of our paper are summarized in three key aspects.

- We introduce a simple and efficient TS-style algorithm, named `LinVDTS` (Algorithm 2), that can utilize variance information for linear contextual bandits. We prove that the *expected* regret of `LinVDTS` is bounded by $\widetilde{\mathbb{O}}\left(d^{3/2}\left(1+\sqrt{\sum_{t=1}^{T}\sigma_t^2}\right)\right)$, where $d$ is the ambient dimension and $\sigma_t^2$ is the variance of the noise in round $t$. We also prove that, *with high probability*, the regret of `LinVDTS` is bounded by $\widetilde{\mathbb{O}}\left(d^{3/2}\left(1+\sqrt{\sum_{t=1}^{T}\sigma_t^2}\right)+\sqrt{T}\right)$. In the analysis of `LinVDTS`, we identify the inherent difficulty of controlling the uncertainty induced by posterior sampling, which precludes a high-probability guarantee that is aware of the noise variance.

- We devise a stratified sampling procedure that fulfills more efficient exploration for linear contextual bandits by exerting explicit uncertainty control over the context vectors. Based on this framework, we propose `LinNATS` (Algorithm 3), a noise-adaptive variant of linear TS. To the best of our knowledge, this is the first noise-adaptive TS algorithm for linear contextual bandits with heteroscedastic noise.

- We prove that `LinNATS` enjoys a high-probability regret bound of order $\widetilde{\mathbb{O}}\left(d^{3/2}\left(1+\sqrt{\sum_{t=1}^{T}\sigma_t^2}\right)\right)$, under standard assumptions for linear contextual bandits with heteroscedastic noise. This improves over the existing $\widetilde{\mathbb{O}}(d^{3/2}\sqrt{T})$ regret bound for linear Thompson sampling (Agrawal and Goyal, 2013; Abeille and Lazaric, 2017), especially when the noise variance diminishes. Our analysis bypasses the standard anti-concentration argument thanks to the stratification framework.

**Notation.** For any $n \in \mathbb{N}^+$, we denote $[n] := \{1, 2, \ldots, n\}$, and $x_{1:n}$ is a shorthand for the set $\{x_1, \ldots, x_n\}$. We write scalars in normal font while representing vectors and matrices in bold font. For any vector $\mathbf{x}$ and positive semi-definite matrix $\mathbf{\Sigma}$, we define $\|\mathbf{x}\|_{\mathbf{\Sigma}} = \|\mathbf{\Sigma}^{1/2}\mathbf{x}\|_2$. For two functions $f$ and $g$ defined on $\mathbb{N}^+$, we use $f(n) \lesssim g(n)$ to indicate that there exists a universal constant $C > 0$ such that $f(n) \leq C \cdot g(n)$ for all $n \in \mathbb{X}$; $f(n) \gtrsim g(n)$ is defined analogously. We use $f(n) \asymp g(n)$ to imply that both $f(n) \lesssim g(n)$ and $f(n) \gtrsim g(n)$ hold true. We use $\mathcal{O}(\cdot)$ to hide constant factors and $\widetilde{\mathcal{O}}(\cdot)$ to further hide poly-logarithmic terms.

## 2 Related work

**Heteroscedastic linear bandits.** The challenge of heteroscedastic noise in linear bandits has been studied through a multitude of perspectives: In particular, Kirschner and Krause (2018) studied linear bandits with heteroscedastic noise via the approach of information-directed sampling. Dai et al. (2022) examined heteroscedastic sparse linear bandits and demonstrated a versatile framework capable of transforming any heteroscedastic linear bandit algorithm into an algorithm tailored for heteroscedastic sparse linear bandits. Building on the UCB approach, Zhou et al. (2021) developed an adaptive algorithm that handles known-variance noise, providing performance guarantees of $\widetilde{\mathcal{O}}\left(\sqrt{dT} + d\sqrt{\sum_{t=1}^{T} \sigma_t^2}\right)$. Zhang et al. (2021) introduced a variance-aware confidence set using elimination with peeling and proved a regret bound of $\widetilde{\mathcal{O}}\left(\text{poly}(d)\sqrt{1 + \sum_{t=1}^{T} \sigma_t^2}\right)$. Kim et al. (2022b) further refined this upper bound to $\widetilde{\mathcal{O}}\left(d^2 + d^{3/2}\sqrt{\sum_{t=1}^{T} \sigma_t^2}\right)$ for linear bandits. Zhao et al. (2022) proposed a multi-layered design for UCB algorithm that achieves a $\widetilde{\mathcal{O}}\left((R+1)\sqrt{dT} + d\sqrt{\sum_{t=1}^{T} \sigma_t^2}\right)$ regret. More recently, Zhao et al. (2023) proposed a UCB-style algorithm with a unified regret upper bound of $\widetilde{\mathcal{O}}\left(d + d\sqrt{\sum_{t=1}^{T} \sigma_t^2}\right)$, where the guarantee is facilitated by the introduction of a novel Freedman-type concentration inequality for self-normalized martingales.

**Thompson sampling.** TS as a method to address the exploration-exploitation trade-off has gained significant attention in recent years due to its simplicity, adaptability, and robust empirical performance (Chapelle and Li, 2011; Gopalan et al., 2014; Kandasamy et al., 2018). Its effectiveness has been demonstrated across various scenarios (Russo et al., 2018), and in particular, a multitude of theoretical results have been solidified within multi-armed bandit settings (May et al., 2012; Kaufmann et al., 2012; Korda et al., 2013; Russo and Van Roy, 2016). For linear contextual bandits, Agrawal and Goyal (2013) provided the first proof for the $\widetilde{\mathcal{O}}(d^{3/2}\sqrt{T})$ regret of linear TS, and later Abeille and Lazaric (2017) delivered an alternate proof and further extended the analysis to more general linear problems. Notably, Hamidi and Bayati (2020) showed that the $\widetilde{\mathcal{O}}(d^{3/2}\sqrt{T})$ rate cannot be improved in worse-case scenarios. While under additional assumptions such as regularity of the contexts, better rates can be achieved (Hamidi and Bayati, 2020; Kim et al., 2021, 2022a).

Moreover, Zhang (2022) introduced a modified version called Feel-Good TS, which is more aggressive in exploring new actions and achieves an $\widetilde{\mathcal{O}}(d\sqrt{T})$ regret. Recently, Luo and Bayati (2023) also proposed a geometry-aware approach, enabling the establishment of a minimax optimal regret of order $\widetilde{\mathcal{O}}(d\sqrt{T})$ for the TS algorithm. Xu et al. (2022) proposed a Langevin Monte Carlo TS algorithm that samples from the posterior distribution beyond Laplacian approximation. Investigations have also been undertaken into TS for kernelized bandits (Chowdhury and Gopalan, 2017) as well as integration of TS with neural networks (Wang and Zhou, 2020; Zhang et al., 2020). Recently, Saha and Kveton (2023) proposed a variance-aware TS algorithm on the topic of Bayesian bandits with a variance-dependent upper bound on its Bayesian regret. As far as our awareness extends, no existing results address noise-adaptive TS algorithms in the context of linear bandits.

## 3 Preliminaries

In this section, we introduce the fundamental framework for contextual linear bandits. We also provide a concise overview of the linear TS algorithm, which functions as our benchmark methodology.

---

**Algorithm 1** Linear Thompson Sampling

---
1: **for** $t = 1, \ldots, T$ **do**
2:      Sample $\boldsymbol{\theta}_t^{\mathrm{TS}}$ from $\mathcal{N}(\widehat{\boldsymbol{\theta}}_t, \beta^2 \mathbf{V}_t^{-1})$
3:      Select arm $\mathbf{x}_t \leftarrow \arg\max_{\mathbf{x} \in \mathcal{X}_t} \langle \mathbf{x}, \boldsymbol{\theta}_t^{\mathrm{TS}} \rangle$ and observe reward $r_t$
4: **end for**

---

## 3.1 Linear contextual bandits

We investigate a linear contextual bandit problem with heteroscedastic noise. Let $T$ be the total number of bandit selection rounds and let $d$ be the ambient dimension. In each round $t \in [T]$, the environment generates an arbitrary set of context vectors $\mathcal{X}_t \subseteq \mathbb{R}^d$, potentially even in an adversarial manner, where each $\mathbf{x} \in \mathcal{X}_t$ denotes the context vector of a feasible action. Upon observing the decision set $\mathcal{X}_t$, the agent selects a context vector $\mathbf{x}_t \in \mathcal{X}_t$, and then receives a reward $r_t$ from the environment. In particular, here the reward is a linear function of the selected context further corrupted by noise, *i.e.*,

$$r_t = \mathbf{x}_t^{\mathsf{T}} \boldsymbol{\theta}^* + \varepsilon_t,$$

where $\boldsymbol{\theta}^* \in \mathbb{R}^d$ represents the true model parameter and $\varepsilon_t$ is the stochastic noise. In this paper, we impose standard assumptions on the linear contextual bandit model, which are common in literature.

**Assumption 3.1.** The ground truth $\boldsymbol{\theta}^*$ satisfies $\|\boldsymbol{\theta}^*\|_2 \leq 1$. For all $t \in [T]$, the decision set $\mathcal{X}_t$ is contained in the unit ball, *i.e.*, $\|\mathbf{x}\|_2 \leq 1$ for all $\mathbf{x} \in \mathcal{X}_t$. There exists a constant $R > 0$ such that $|\varepsilon_t| \leq R$ for all $t \in [T]$. For every $t \in [T]$, $\mathbb{E}[\varepsilon_t \mid \mathbf{x}_{1:t}, \varepsilon_{1:t-1}] = 0$ and $\mathbb{E}[\varepsilon_t^2 \mid \mathbf{x}_{1:t}, \varepsilon_{1:t-1}] = \sigma_t^2$.

In each round $t$, an algorithm selects an arm $\mathbf{x}_t \in \mathcal{X}_t$, and we denote the optimal arm as $\mathbf{x}_t^*$, *i.e.*, $\mathbf{x}_t^* = \arg\max_{\mathbf{x} \in \mathcal{X}_t} \mathbf{x}^{\mathsf{T}} \boldsymbol{\theta}^*$. As a result, the suboptimality of the selected arm at time $t$ can be expressed as $\Delta_t = \mathbf{x}_t^{*\mathsf{T}} \boldsymbol{\theta}^* - \mathbf{x}_t^{\mathsf{T}} \boldsymbol{\theta}^*$. The objective of the agent is to minimize the cumulative regret incurred over the time horizon $T$, which is defined as

$$\mathcal{R}(T) = \sum\nolimits_{t=1}^{T} \Delta_t = \sum\nolimits_{t=1}^{T} \langle \mathbf{x}_t^* - \mathbf{x}_t, \boldsymbol{\theta}^* \rangle.$$

## 3.2 Thompson sampling for linear contextual bandits

A standard TS algorithm for linear contextual bandits (*i.e.*, Algorithm 1) was first proposed in Agrawal and Goyal (2013), employing Gaussian priors and likelihood functions in the design. At each round $t \in [T]$, a sample $\boldsymbol{\theta}_t^{\mathrm{TS}}$ is drawn from the Gaussian posterior distribution, centered at the estimate $\widehat{\boldsymbol{\theta}}_t = \mathbf{V}_t^{-1} \sum_{s=1}^{t-1} r_s \mathbf{x}_s$ with covariance matrix $\beta^2 \mathbf{V}_t^{-1}$, where $\mathbf{V}_t = \mathbf{I}_d + \sum_{s=1}^{t-1} \mathbf{x}_s \mathbf{x}_s^{\mathsf{T}}$ and the confidence radius $\beta = 3R\sqrt{d \log(T/\delta)}$. The arm $\mathbf{x}_t$ is then selected against the sample $\boldsymbol{\theta}_t^{\mathrm{TS}}$.

The main idea here is to achieve exploitation by updating the posterior using collected information, and the exploration among the arms is performed naturally in posterior sampling. Agrawal and Goyal (2013) demonstrated that the algorithm presented in Algorithm 1 achieves a regret bound of order $\widetilde{\mathcal{O}}(d^{3/2}\sqrt{T})$. This bound is worse than the minimax lower bound by an unavoidable factor of $\sqrt{d}$ (Hamidi and Bayati, 2020). Despite this discrepancy, the regret bound effectively showcases the algorithm's ability to balance exploration and exploitation in the linear contextual bandit setting.

# 4 Warm up: a simple and variance-dependent Thompson sampling algorithm

In this section, we introduce a preliminary algorithm, LinVDTS, delineated in Algorithm 2, which combines TS with a weighted ridge regression estimator in a straightforward manner. For the sake of clear exposition, we focus on a less complex linear bandit problem in which the agent is privy to both the reward $r_t$ and its variance $\sigma_t^2$ subsequent to the selection of an arm.

## 4.1 Algorithmic design

In order to integrate variance information into the estimation process, the use of weighted regression has emerged as a prevalent approach in the field of variance-aware online learning (Zhou et al., 2021;

**Algorithm 2 Lin**ear **V**ariance-**D**ependent **T**hompson **S**ampling (`LinVDTS`)

---

**Require:** Total number of iterations $T$, number of total arms $n$
1: Initialize $\mathbf{V}_1 \leftarrow \lambda \mathbf{I}_d$, $\boldsymbol{\nu}_1 \leftarrow 0$, $\widehat{\boldsymbol{\theta}}_1 \leftarrow 0$, $\widehat{\beta}_1 \leftarrow \sqrt{\lambda}$, and $\boldsymbol{\Sigma}_1 \leftarrow \widehat{\beta}_1^2 \mathbf{V}_1^{-1}$
2: **for** $t = 1, \ldots, T$ **do**
3:     Observe context vectors $\mathcal{X}_t$
4:     Draw $\boldsymbol{\theta}_t^{\mathrm{TS}} \sim \mathcal{N}(\widehat{\boldsymbol{\theta}}_t, \boldsymbol{\Sigma}_t)$
5:     $\mathbf{x}_t \leftarrow \arg\max_{\mathbf{x} \in \mathcal{X}_t} \langle \mathbf{x}, \boldsymbol{\theta}_t^{\mathrm{TS}} \rangle$
6:     Observe reward $r_t$ and variance $\sigma_t^2$
7:     Update $\mathbf{V}_{t+1} \leftarrow \mathbf{V}_t + \mathbf{x}_t \mathbf{x}_t^{\mathsf{T}} / \overline{\sigma}_t^2$
8:     Update $\boldsymbol{\nu}_{t+1} \leftarrow \boldsymbol{\nu}_t + r_t \mathbf{x}_t / \overline{\sigma}_t^2$
9:     Update $\widehat{\boldsymbol{\theta}}_{t+1} \leftarrow \mathbf{V}_{t+1}^{-1} \boldsymbol{\nu}_{t+1}$
10:    Compute covariance matrix $\boldsymbol{\Sigma}_{t+1} \leftarrow \widehat{\beta}_{t+1}^2 \mathbf{V}_{t+1}^{-1}$
11: **end for**

---

Min et al., 2021b, 2022a; Zhou and Gu, 2022; Yin et al., 2022; Zhao et al., 2023). This methodology is intimately associated with the concept of the Best Linear Unbiased Estimator (BLUE), a well-established statistical technique for achieving optimal linear estimation (Henderson, 1975).

Let us first explain the details of the proposed algorithm `LinVDTS`. In every round $t$, the algorithm computes an estimate $\widehat{\boldsymbol{\theta}}_t$ of the ground truth $\boldsymbol{\theta}^*$ by solving a weighted ridge regression problem:

$$\widehat{\boldsymbol{\theta}}_t = \arg\min_{\boldsymbol{\theta}} \sum_{s=1}^{t} \frac{1}{\overline{\sigma}_s^2} (r_s - \langle \mathbf{x}_s, \boldsymbol{\theta} \rangle)^2 + \lambda \|\boldsymbol{\theta}\|_2^2.$$

Here $\overline{\sigma}_t$ represents a weight parameter and $\lambda$ denotes a regularization constant. For each approximation $\widehat{\boldsymbol{\theta}}_t$, we stochastically draw $\boldsymbol{\theta}_t^{\mathrm{TS}}$ from a Gaussian distribution with mean $\widehat{\boldsymbol{\theta}}_t$ and covariance $\boldsymbol{\Sigma}_t = \widehat{\beta}_t \mathbf{V}_t^{-1}$, where $\mathbf{V}_t = \lambda \mathbf{I}_d + \sum_{s=1}^{t-1} \mathbf{x}_s \mathbf{x}_s^{\mathsf{T}} / \overline{\sigma}_s^2$ is computed from previous context and reward pairs, and the confidence radius, $\widehat{\beta}_t$, is chosen as

$$\widehat{\beta}_t \asymp \sqrt{d \log\left(\frac{t}{d\lambda\alpha^2}\right) \log\left(\frac{t^2}{\delta} \log \frac{\gamma^2}{\alpha}\right)} + \frac{R}{\gamma^2} \log\left(\frac{t^2}{\delta} \log \frac{\gamma^2}{\alpha}\right) + \sqrt{\lambda}.$$

Refer to (A.2) for a detailed delineation of $\widehat{\beta}_t$. For each context vector $\mathbf{x} \in \mathcal{X}_t$, we compute the estimated reward as $\langle \mathbf{x}, \boldsymbol{\theta}_t^{\mathrm{TS}} \rangle$, and the algorithm outlined in Algorithm 2 then selects the arm $\mathbf{x}_t$ that optimizes this estimated reward. Following the approach of Zhou and Gu (2022), we construct the weight parameter $\overline{\sigma}_t = \max\{\sigma_t, \alpha, \gamma \|\mathbf{x}_t\|_{\mathbf{V}_t^{-1}}^{1/2}\}$, which is designed as the maximum among the variance, a constant $\alpha$, and the uncertainty associated with the selected arm $\mathbf{x}_t$. It ensures the formulation of a tight confidence radius $\widehat{\beta}_t$, guided by a Bernstein-type concentration inequality, *cf.* Theorem A.1.

## 4.2 Regret guarantee and technical challenges

The theorem below establishes a variance-dependent upper bound on the expected regret of `LinVDTS`. See Appendix A.3 for detailed proof.

**Theorem 4.1.** *Set parameters* $\alpha = 1/\sqrt{T}$, $\gamma = R^{\frac{1}{2}}/d^{\frac{1}{4}}$, $\lambda = d$. *For any* $\delta \in (0, 1)$, *the expected regret of Algorithm 2 is upper bounded as follows:*

$$\mathbb{E}[\mathcal{R}(T)] = \mathcal{O}\left(d^{3/2}\left(1 + \sqrt{\sum_{t=1}^{T} \sigma_t^2}\right) \log T + \delta T\right).$$

*Further, it holds with probability* $1 - \delta$ *that*

$$\mathcal{R}(T) = \mathcal{O}\left(d^{3/2}\left(1 + \sqrt{\sum_{t=1}^{T} \sigma_t^2}\right) \log T + \sqrt{T \log \frac{1}{\delta}}\right).$$

*Remark* 4.2. Choosing $\delta = 1/T$, then we see that the expected regret of Algorithm 2 is bounded by $\widetilde{\mathcal{O}}\left(d^{3/2}\left(1 + \sqrt{\sum_{t=1}^{T}\sigma_t^2}\right)\right)$. However, the high-probability regret bound of Algorithm 2 is of order $\widetilde{\mathcal{O}}\left(d^{3/2}\left(1 + \sqrt{\sum_{t=1}^{T}\sigma_t^2}\right) + \sqrt{T}\right)$, where the additional $\sqrt{T}$ term fails to be variance-dependent. Nonetheless, this regret bound already improves over the existing $\widetilde{\mathcal{O}}(d^{3/2}\sqrt{T})$ bound for linear Thompson sampling in the literature (Agrawal and Goyal, 2013; Abeille and Lazaric, 2017).

The limitation on the high-probability bound arises from a set of technical challenges that are implicit and inherent in the posterior sampling used in Algorithm 2. To illustrate this, we now provide an outline of the proof for Theorem 4.1, and then discuss the technical intricacies therein.

*Proof sketch of Theorem 4.1.* Let us denote $\Delta_t(\mathbf{x}) = \langle \mathbf{x}_t^* - \mathbf{x}, \boldsymbol{\theta}^* \rangle$ as the suboptimality of $\mathbf{x} \in \mathcal{X}_t$. We introduce the saturated set $\mathcal{S}(t)$ for round $t$ as the ensemble of arms whose confidence radius is dwarfed by its suboptimality, *i.e.*, $\mathcal{S}(t) = \left\{\mathbf{x} \in \mathcal{X}_t : \Delta_t(\mathbf{x}) > (2\sqrt{d\log t} + 1)\widehat{\beta}_t\|\mathbf{x}\|_{\mathbf{V}_t^{-1}}\right\}$ (Agrawal and Goyal, 2013). Then the suboptimality $\Delta_t$ of the chosen context $\mathbf{x}_t$ can be decomposed as follows:

$$\Delta_t = \langle \mathbf{x}_t^* - \mathbf{x}_t, \boldsymbol{\theta}^* \rangle = \Delta_t(\mathbf{x}_t^\dagger) + \mathbf{x}_t^{\dagger\mathsf{T}}\boldsymbol{\theta}^* - \mathbf{x}_t^\mathsf{T}\boldsymbol{\theta}^*,$$

where the term $\mathbf{x}_t^\dagger = \arg\min_{\mathbf{x}\notin\mathcal{S}(t)}\|\mathbf{x}\|_{\mathbf{V}_t^{-1}}$ refers to the unsaturated arm that possesses the smallest $\mathbf{V}_t^{-1}$ norm. Leveraging the concentration results delineated in Lemmata A.2 and A.3 pertaining to $\widehat{\boldsymbol{\theta}}_t$ and $\boldsymbol{\theta}_t^{\mathrm{TS}}$, we have

$$\Delta_t \lesssim \sqrt{d\log t}(\|\mathbf{x}_t^\dagger\|_{\mathbf{V}_t^{-1}} + \|\mathbf{x}_t\|_{\mathbf{V}_t^{-1}})\widehat{\beta}_t. \tag{4.1}$$

For upper bound on the expected regret, we use an anti-concentration argument (Lemma A.6) to relate $\|\mathbf{x}_t^\dagger\|_{\mathbf{V}_t^{-1}}$ back to $\|\mathbf{x}_t\|_{\mathbf{V}_t^{-1}}$. This yields

$$\mathbb{E}[\Delta_t \mid \mathscr{H}_t] \lesssim \sqrt{d\log t}\,\mathbb{E}[\|\mathbf{x}_t\|_{\mathbf{V}_t^{-1}}]\widehat{\beta}_t + \frac{2}{t^2}, \tag{4.2}$$

where $\{\mathscr{H}_t\}_{t\geq 1}$ represents a filtration of the information available up to the observation of the set of context vectors at each round. Then collecting the above inequality for all $t \in [T]$, together with an elliptical potential lemma, we get the desired upper bound for $\mathbb{E}[\mathcal{R}(T)]$.

Next, for the high-probability regret bound, we decompose

$$\mathcal{R}(T) = \sum_{t=1}^{T}\mathbb{E}[\Delta_t \mid \mathscr{H}_t] + \sum_{t=1}^{T}(\Delta_t - \mathbb{E}[\Delta_t \mid \mathscr{H}_t]) \tag{4.3}$$

where the first term corresponds to the expected regret bound. We control the second term using martingale concentration, which results in an additional $\sqrt{T}$ term in the final regret bound. □

## Why cannot simultaneously achieve variance awareness and high-probability bound?

First, the optimism induced by the posterior sampling is too conservative.[1] A primary challenge in the analysis is to effectively control each suboptimality term $\Delta_t$, *cf.* (4.1). It is important to note that one key algorithmic design in UCB-type algorithms is the exact optimism from the UCB bonus $\beta\|\mathbf{x}_t\|_{\mathbf{V}_t^{-1}}$, which ensures $\langle x_t^*, \widehat{\boldsymbol{\theta}}_t\rangle + \beta\|\mathbf{x}_t^*\|_{\mathbf{V}_t^{-1}} \leq \langle x_t, \widehat{\boldsymbol{\theta}}_t\rangle + \beta\|\mathbf{x}_t\|_{\mathbf{V}_t^{-1}}$. However, for TS-style algorithms, the absence of this exact optimism hinders our control over the term $\|\mathbf{x}_t^*\|_{\mathbf{V}_t^{-1}}$ throughout all rounds $t \in [T]$, *i.e.*, a standard decomposition of $\Delta_t = \langle \mathbf{x}_t^* - \mathbf{x}_t, \boldsymbol{\theta}^*\rangle$ yields $\Delta_t \leq (2\sqrt{d\log t} + 1)\widehat{\beta}_t(2\|\mathbf{x}_t^*\|_{\mathbf{V}_t^{-1}} + \|\mathbf{x}_t\|_{\mathbf{V}_t^{-1}})$ in this case. As a result, standard analyses turn to methodologies that employ an anti-concentration inequality on the sampling distribution (Agrawal and Goyal, 2013). This leads to the inequality $\|\mathbf{x}_t^\dagger\|_{\mathbf{V}_t^{-1}} \leq \|\mathbf{x}_t\|_{\mathbf{V}_t^{-1}}$ with a constant probability, thereby providing an upper bound on the expected regret.

Second, in the conversion of the expected regret bound to a high-probability bound, we have no help from the noise variance information. One may wonder if it is possible to incorporate the

---

[1] This is also identified in Zhang (2022) as the source of suboptimal dependence on dimension in the regret bound for TS.

variance of $\Delta_t - \mathbb{E}[\Delta_t \mid \mathscr{H}_t]$ to get a Bernstein-type concentration result. However, the variance of $\Delta_t - \mathbb{E}[\Delta_t \mid \mathscr{H}_t]$ does not necessarily conform to the noise variance, especially when the decision set $\mathfrak{X}_t$ is revealed adversarially. To see this, consider the following situation: Given $\widehat{\boldsymbol{\theta}}_t$, the environment reveals a decision set $\mathscr{X}_t = \{\mathbf{x}_\perp, -\mathbf{x}_\perp\}$ where $\mathbf{x}_\perp$ is orthogonal to $\widehat{\boldsymbol{\theta}}_t$. Then a small perturbation in $\boldsymbol{\theta}_t^{\text{TS}}$ along the direction of $\mathbf{x}_\perp$ can cause the change of arm selection. This implies that a small uncertainty of the posterior distribution along the direction of $\mathbf{x}_\perp$ will be amplified to a constant variation of the selected arm $\mathbf{x}_t$. Therefore, in general, we do not have any delicate control of the variance of $\Delta_t - \mathbb{E}[\Delta_t \mid \mathscr{H}_t]$. Consequently, we can only apply Azuma-Hoeffding inequality to get a non-noise-adaptive term of $\widetilde{\mathbb{O}}(\sqrt{T})$. We discuss in the next section how to conquer this via more advanced algorithmic design.

# 5 General case: a noise-adaptive Thompson sampling algorithm

In this section, we propose another TS algorithm `LinNATS` for linear contextual bandits, one that is provably adaptive to unknown heteroscedastic noise. For greater generality, we adopt the identical problem setting delineated in Section 3, and we refrain from assuming access to the variance information $\{\sigma_t^2\}_{t=1}^T$ associated with the rewards $\{r_t\}_{t=1}^T$.

## 5.1 Algorithm

The proposed algorithm `LinNATS` is displayed in Algorithm 3. Below we go through the details of `LinNATS` and explain the algorithmic design along the way.

**Stratification over contextual uncertainty.** For more efficient uncertainty control, we adopt a stratification strategy akin to that previously used for UCB-type algorithms (Chu et al., 2011; Li et al., 2023; Zhao et al., 2023). Our algorithm segments the context vectors at each round $t$ into $L$ distinct layers, enabling the precise control of both $\|\mathbf{x}_t\|_{\mathbf{V}_t^{-1}}$ and $\|\mathbf{x}_t^*\|_{\mathbf{V}_t^{-1}}$.

Specifically, the $\ell$-th layer establishes a threshold of $2^{-\ell}$ and retains a separate estimate $\widehat{\boldsymbol{\theta}}_{t,\ell}$ and sampled variable $\boldsymbol{\theta}_{t,\ell}^{\text{TS}}$ for each $\ell \in [L]$ and $t \in [T]$ (Line 3). Commencing from $\ell = 1$, a sequence of decision sets $\mathfrak{X}_t = \mathfrak{X}_{t,1} \supseteq \mathfrak{X}_{t,2} \supseteq \dots$ of diminishing sizes is derived by excluding the arms $\mathbf{x} \in \mathfrak{X}_{t,\ell}$ that are less likely to maximize $\langle \mathbf{x}, \boldsymbol{\theta}^* \rangle$ at each layer $\ell$. More specifically, on Line 11 we let

$$\mathfrak{X}_{t,\ell+1} = \left\{ \mathbf{x} \in \mathfrak{X}_{t,\ell} : \langle \mathbf{x}, \boldsymbol{\theta}_{t,\ell}^{\text{TS}} \rangle \geq \max_{\mathbf{x}' \in \mathfrak{X}_{t,\ell}} \langle \mathbf{x}', \boldsymbol{\theta}_{t,\ell}^{\text{TS}} \rangle - 2^{-\ell+1} (\sqrt{2d\log(4t^2 L/\delta)} + 1)\widehat{\beta}_{t,\ell} \right\}, \quad (5.1)$$

which ensures that the optimal arm $\mathbf{x}_t^*$ falls into all decision sets with high probability. The context vector $\mathbf{x}_t = \arg\max_{\mathbf{x} \in \mathfrak{X}_{t,\ell}} \langle \mathbf{x}, \boldsymbol{\theta}_{t,\ell}^{\text{TS}} \rangle$ is chosen if the uncertainty term $\|\mathbf{x}\|_{\mathbf{V}_{t,\ell}^{-1}} \leq \alpha$ for all $\mathbf{x} \in \mathfrak{X}_{t,\ell}$ (Line 8). Otherwise, $\mathbf{x}_t$ is selected only when the uncertainty surpasses the threshold, i.e., $\|\mathbf{x}_t\|_{\mathbf{V}_{t,\ell}^{-1}} \geq 2^{-\ell}$ for some $\ell$, and the elimination process in (5.1) terminates (Line 13).

The round index $t$ is incorporated into a growing index set $\Psi_{t+1,\ell}$ (Line 9 & 15), and the observation pair $(\mathbf{x}_t, r_t)$ participates the later estimation of $\widehat{\boldsymbol{\theta}}_{s,\ell}$ for all $s > t$ only if the chosen arm exhibits large uncertainty within the current layer, i.e., $\|\mathbf{x}_t\|_{\mathbf{V}_{t,\ell}^{-1}} \geq 2^{-\ell}$ (Line 21). This process guarantees that every context vector $\mathbf{x} \in \mathfrak{X}_{t,\ell}$, cascading through the first $(\ell-1)$-th layer, satisfies $\|\mathbf{x}\|_{\mathbf{V}_{t,\ell-1}^{-1}} \leq 2^{-\ell+1}$.

**Parameter estimation with weighted ridge regression.** To estimate $\boldsymbol{\theta}^*$, we again utilize weighted ridge regression, but this time within each uncertainty level. For each layer $\ell \in [L]$ and round $t \in [T]$, the associated estimator $\widehat{\boldsymbol{\theta}}_{t,\ell}$ is given by

$$\widehat{\boldsymbol{\theta}}_{t,\ell} = \arg\min_{\boldsymbol{\theta} \in \mathbb{R}^d} \sum_{s \in \Psi_{t,\ell}} w_s^2 (r_s - \langle \mathbf{x}_s, \boldsymbol{\theta} \rangle)^2 + 2^{-2\ell} \|\boldsymbol{\theta}\|_2^2,$$

where the weight parameter $w_t > 0$ is selected to fulfill the condition $\|w_t \mathbf{x}_t\|_{\mathbf{V}_{t,\ell}^{-1}} = 2^{-\ell}$ (Line 14). Owing to the specific formulation of the weighting parameter, it follows that $\sup_{s \in \Psi_{t,\ell}} \|w_s \mathbf{x}_s\|_{\mathbf{V}_{s,\ell}^{-1}} = 2^{-\ell}$ holds universally for all $t \in [T]$ and $\ell \in [L]$. This allows the application of a Freedman-type

**Algorithm 3** **Lin**ear **N**oise-**A**daptive **T**hompson **S**ampling (`LinNATS`)

---

**Require:** Time horizon $T$, number of arms $n$, and threshold $\alpha = T^{-3/2}/R$

1: Initialize $L \leftarrow \lceil \log_2(1/\alpha) \rceil$, $\mathbf{V}_{1,\ell} \leftarrow 2^{-2\ell}\mathbf{I}_d$, $\boldsymbol{\nu}_{1,\ell} \leftarrow 0$, $\widehat{\boldsymbol{\theta}}_{1,\ell} \leftarrow 0$, $\Psi_{1,\ell} \leftarrow \emptyset$ and $\widehat{\beta}_{1,\ell} \leftarrow 2^{-\ell+1}$
    for all $\ell \in [L]$
2: **for** $t = 1, \ldots, T$ **do**
3:     Draw $\boldsymbol{\theta}_{t,\ell}^{\mathrm{TS}} \sim \mathcal{N}(\widehat{\boldsymbol{\theta}}_{t,\ell}, \boldsymbol{\Sigma}_{t,\ell})$ for all $\ell \in [L]$
4:     Observe context vectors $\mathcal{X}_t = (\mathbf{x}_{t,1}, \ldots, \mathbf{x}_{t,n})$
5:     Let $\ell \leftarrow 1$, $\mathcal{X}_{t,\ell} \leftarrow \mathcal{X}_t$
6:     **while** $\mathbf{x}_t$ is *not* selected **do**
7:         **if** $\|\mathbf{x}\|_{\mathbf{V}_{t,\ell}^{-1}} \leq \alpha$ for all $\mathbf{x} \in \mathcal{X}_{t,\ell}$ **then**
8:             Choose $\mathbf{x}_t \leftarrow \arg\max_{\mathbf{x}\in\mathcal{X}_{t,\ell}}\langle \mathbf{x}, \boldsymbol{\theta}_{t,\ell}^{\mathrm{TS}}\rangle$ and observe reward $r_t$
9:             Update $\Psi_{t+1,\ell'} \leftarrow \Psi_{t,\ell'}$ for all $\ell' \in [L]$
10:        **else if** $\|\mathbf{x}\|_{\mathbf{V}_{t,\ell}^{-1}} \leq 2^{-\ell}$ for all $\mathbf{x} \in \mathcal{X}_{t,\ell}$ **then**
11:             Update $\mathcal{X}_{t,\ell+1}$ following (5.1)
12:        **else**
13:             Choose any $\mathbf{x}_t \in \mathcal{X}_{t,\ell}$ such that $\|\mathbf{x}_t\|_{\mathbf{V}_{t,\ell}^{-1}} > 2^{-\ell}$ and observe reward $r_t$
14:             Compute $w_t \leftarrow 2^{-\ell}/\|\mathbf{x}_t\|_{\mathbf{V}_{t,\ell}^{-1}}$
15:             Update $\Psi_{t+1,\ell} \leftarrow \Psi_{t,\ell} \cup \{t\}$ and $\Psi_{t+1,\ell'} \leftarrow \Psi_{t,\ell'}$ for all $\ell' \in [L]\backslash\{\ell\}$
16:        **end if**
17:        Update $\ell \leftarrow \ell + 1$
18:     **end while**
19:     **for** $\ell \in [L]$ **do**
20:        **if** $\Psi_{t+1,\ell} \neq \Psi_{t,\ell}$ **then**
21:             Update $\mathbf{V}_{t+1,\ell} \leftarrow \mathbf{V}_{t,\ell} + w_t^2 \mathbf{x}_t\mathbf{x}_t^{\mathsf{T}}$,    $\boldsymbol{\nu}_{t+1,\ell} \leftarrow \boldsymbol{\nu}_{t,\ell} + w_t^2 r_t \mathbf{x}_t$
22:             Update $\widehat{\boldsymbol{\theta}}_{t+1,\ell} \leftarrow \mathbf{V}_{t+1,\ell}^{-1}\boldsymbol{\nu}_{t+1,\ell}$
23:             Compute confidence radius $\widehat{\beta}_{t+1,\ell}$ following Eq. (5.3)
24:             Compute covariance matrix $\boldsymbol{\Sigma}_{t+1,\ell} \leftarrow \widehat{\beta}_{t+1,\ell}^2 \mathbf{V}_{t+1,\ell}^{-1}$
25:        **else**
26:             $\mathbf{V}_{t+1,\ell} \leftarrow \mathbf{V}_{t,\ell}, \boldsymbol{\nu}_{t+1,\ell} \leftarrow \boldsymbol{\nu}_{t,\ell}, \widehat{\boldsymbol{\theta}}_{t+1,\ell} \leftarrow \widehat{\boldsymbol{\theta}}_{t,\ell}, \widehat{\beta}_{t+1,\ell} \leftarrow \widehat{\beta}_{t,\ell}, \boldsymbol{\Sigma}_{t+1,\ell} \leftarrow \boldsymbol{\Sigma}_{t,\ell}$
27:        **end if**
28:     **end for**
29: **end for**

---

concentration inequality to guarantee with high probability for all layer $\ell \in [L]$ that

$$\|\widehat{\boldsymbol{\theta}}_{t,\ell} - \boldsymbol{\theta}^*\|_{\mathbf{V}_{t,\ell}} = \widetilde{\mathcal{O}}\left(\frac{R}{2^\ell} + \frac{1}{2^\ell}\sqrt{\sum_{s\in\Psi_{t,\ell}} w_s^2\sigma_s^2}\right). \tag{5.2}$$

**Variance-dependent confidence radius.** The variance-dependent error bound (5.2) is sufficient for the formulation of a TS confidence radius, provided that the variance $\sigma_t^2$ is known. However, under scenarios where the variance is unknown, it is necessary to further estimate them adaptively and on the fly. Specifically, we estimate the summation of past weighted variances, represented as $\sum_{s\in\Psi_{t,\ell}} w_s^2\sigma_s^2$, each using an empirical estimator $(r_t - \langle\mathbf{x}_t, \widehat{\boldsymbol{\theta}}_{t,\ell}\rangle)^2$ as a substitute for $\sigma_t^2$. The weighted summation of these estimators $\zeta_{t,\ell} = \sum_{s\in\Psi_{t,\ell}} w_s^2(r_s - \langle\mathbf{x}_s, \widehat{\boldsymbol{\theta}}_{s,\ell}\rangle)^2$ effectively acts as a precise estimator of $\sum_{s\in\Psi_{t,\ell}} w_s^2\sigma_s^2$ (Zhao et al., 2023). Utilizing this estimator, we adjust the posterior distribution according to the following confidence radius

$$\widehat{\beta}_{t,\ell} = \frac{1}{2^{\ell-4}}\sqrt{\left(8\zeta_{t,\ell} + 6R^2\log\frac{8t^2L}{\delta} + 2^{-2\ell+4}\right)\log\frac{8t^2L}{\delta}} + \frac{3R}{2^{\ell-1}}\log\frac{8t^2L}{\delta} + \frac{1}{2^{\ell-1}}, \tag{5.3}$$

where the variance estimator takes a slight variant for technical considerations:

$$\zeta_{t,\ell} = \begin{cases} \sum_{s\in\Psi_{t,\ell}} w_s^2(r_s - \langle\mathbf{x}_s, \widehat{\boldsymbol{\theta}}_{t,\ell}\rangle)^2, & \text{if } 2^\ell \geq 64\sqrt{\log(8t^2L/\delta)} \\ R^2|\Psi_{t,\ell}|, & \text{otherwise.} \end{cases} \tag{5.4}$$

## 5.2 Regret analysis

We provide the regret guarantee of Algorithm 3 in the following theorem. We also present a proof sketch of Theorem 5.1 to emphasize the main idea. A complete proof is given in Appendix B.3.

**Theorem 5.1.** *Set parameters $\alpha = T^{-3/2}/R$ and $L = \lceil \log_2(1/\alpha) \rceil$. For any $\delta \in (0, 1)$, choose the confidence radii $\{\widehat{\beta}_{t,\ell}\}_{t\in[T],\ell\in[L]}$ and the variance estimators $\{\zeta_{t,\ell}\}_{t\in[T],\ell\in[L]}$ according to (5.3) and (5.4) respectively. Then with probability at least $1 - 2\delta$, Algorithm 3 has regret guarantee*

$$\mathcal{R}(T) = \mathcal{O}\left( d^{3/2} \left( 1 + R + \sqrt{\sum_{t=1}^{T} \sigma_t^2} \right) \log T \right).$$

*Remark* 5.2. Theorem 5.1 shows that our proposed algorithm, `LinNATS`, delivers a variance-dependent regret guarantee free of any prior knowledge of the noise variance. The regret upper bound is also stipulated to be of the order $\widetilde{\mathcal{O}}\left( d^{3/2} \left( 1 + \sqrt{\sum_{t=1}^{T} \sigma_t^2} \right) \right)$, considering $R$ as a constant. Notably, in scenarios where the variance is constant, our `LinNATS` algorithm recovers the regret guarantee of $\widetilde{\mathcal{O}}(d^{3/2}\sqrt{T})$ for linear TS (Agrawal and Goyal, 2013; Abeille and Lazaric, 2017), and in situations where the reward is deterministic, it attains a regret guarantee of $\widetilde{\mathcal{O}}(d^{3/2})$. This highlights the versatility and adaptive performance of our proposed algorithm across various environmental settings.

Next, we present the main ingredients of our analysis for Algorithm 3.

*Proof sketch of Theorem 5.1.* The choice of the weight parameters in `LinNATS` allows us to first apply a standard Freedman-type concentration argument (Theorem B.1), and deduce in Lemma B.2 that the estimator $\widehat{\boldsymbol{\theta}}_{t,\ell}$ is accurate up to a variance-dependent quantity $\widetilde{\beta}_{t,\ell}$:

$$\|\widehat{\boldsymbol{\theta}}_{t,\ell} - \boldsymbol{\theta}^*\|_{\mathbf{V}_{t,\ell}} \le \widetilde{\beta}_{t,\ell} \asymp \frac{1}{2^{\ell}} \sqrt{\sum_{s \in \Psi_{t,\ell}} w_s^2 \sigma_s^2 \log \frac{t^2 L}{\delta}} + \frac{R}{2^{\ell}} \log \frac{t^2 L}{\delta} + \frac{1}{2^{\ell-1}}.$$

However, $\widetilde{\beta}_{t,\ell}$ comprises the variance information $\{\sigma_s^2\}_{s \in \Psi_{t,\ell}}$, which is unknown to the algorithm and thus cannot be employed in the construction of the TS process. Thanks to Lemma B.4, the sum of past weighted variances $\sum_{s \in \Psi_{t,\ell}} w_s^2 \sigma_s^2$ can be upper-bounded by the empirical estimator $\zeta_{t,\ell}$ defined in (5.4). Therefore, we can use $\widehat{\beta}_{t,\ell}$ as defined in (5.3) for the confidence radius, and show that

$$\|\boldsymbol{\theta}_{t,\ell}^{\text{TS}} - \boldsymbol{\theta}^*\|_{\mathbf{V}_{t,\ell}} \le (\sqrt{2d \log(4t^2 L/\delta)} + 1)\widehat{\beta}_{t,\ell}.$$

Then, by leveraging the stratified structure in conjunction with the cascading arm selection procedure across $\ell \in [L]$, we are equipped to exert uniform control over $\|\mathbf{x}\|_{\mathbf{V}_{t,\ell-1}^{-1}}$ for every $\mathbf{x} \in \mathcal{X}_{t,\ell}$ in each instance of $t \in \Psi_{T+1,\ell}$, which resolves the technical challenge laid out in Section 4.2. Note that for any context vector $\mathbf{x} \in \mathcal{X}_{t,\ell}$, we have $\|\mathbf{x}\|_{\mathbf{V}_{t,\ell-1}^{-1}} \le 2^{-\ell+1}$, and thus by Cauchy-Schwarz inequality

$$|\langle \mathbf{x}, \boldsymbol{\theta}_{t,\ell}^{\text{TS}} - \boldsymbol{\theta}^* \rangle| \le \|\mathbf{x}\|_{\mathbf{V}_{t,\ell-1}^{-1}} \|\boldsymbol{\theta}_{t,\ell}^{\text{TS}} - \boldsymbol{\theta}^*\|_{\mathbf{V}_{t,\ell-1}} \le 2^{-\ell+1}(\sqrt{2d \log(4t^2 L/\delta)} + 1)\widehat{\beta}_{t,\ell}. \quad (5.5)$$

Indeed, we ascertain in Lemma B.5 via an induction argument that for all $t \ge 1$ and $\ell \in [L]$, where $\mathcal{X}_{t,\ell}$ exists, $\mathbf{x}_t^* \in \mathcal{X}_{t,\ell}$. Hence (5.5) holds for both $\mathbf{x}_t^*$ and $\mathbf{x}_t$ as a consequence of the stratified structure. This allows us to execute the following decomposition of suboptimality for all $t \in \Psi_{T+1,\ell}$:

$$\Delta_t \le \underbrace{\langle \mathbf{x}_t^*, \boldsymbol{\theta}_{t,\ell-1}^{\text{TS}} \rangle - \langle \mathbf{x}_t, \boldsymbol{\theta}_{t,\ell-1}^{\text{TS}} \rangle}_{\le 0 \text{ by the definition of } \mathbf{x}_t} + \underbrace{|\langle \mathbf{x}_t^*, \boldsymbol{\theta}_{t,\ell-1}^{\text{TS}} - \boldsymbol{\theta}^* \rangle| + |\langle \mathbf{x}_t, \boldsymbol{\theta}_{t,\ell-1}^{\text{TS}} - \boldsymbol{\theta}^* \rangle|}_{\text{controlled via (5.5)}}$$

$$\le 8 \cdot 2^{-\ell}(\sqrt{2d \log(4t^2 L/\delta)} + 1)\widehat{\beta}_{t,\ell-1}.$$

The main result is then obtained by collating the preceding inequalities for all $\ell \in [L]$ followed by an elliptical potential inequality, taking into account that $L = \mathcal{O}(\log T)$. $\qquad\square$

# 6 Conclusion and future work

In this paper, we aimed to address the task of noise-adaptive learning on linear contextual bandits with indeterminate heteroscedastic variance. As part of our endeavor, we put forth a straightforward algorithm, `LinVDTS`, which assures a variance-dependent guarantee on the expected regret. Our analytical exploration reveals that a simplistic implementation of Thompson Sampling culminates in a sub-optimal regret bound. To mitigate this issue, we put forth an innovative Thompson Sampling algorithm, `LinNATS`. With the incorporation of a stratification scheme, the algorithm successfully navigates the technical challenges of uncertainty control, securing a variance-dependent regret under unknown variance. Consequently, it effectively bridges the performance chasm between the worst-case scenarios involving constant variance and those concerning deterministic rewards.

Looking forward, it would be intriguing to conceive a noise-adaptive variant of the recently devised Feel-Good TS as introduced by Zhang (2022), which enhances the dependency on the dimension $d$. Furthermore, extending the noise-adaptive methodology to TS algorithms designed for more general settings, *e.g.*, generalized linear bandits and Reinforcement Learning with linear function approximation, constitutes an appealing direction for future research.

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

# A  Regret analysis of Algorithm 2

## A.1  Clarification of notation

Before elaborating on proofs, we delineate several shorthands to facilitate our discourse. For each $t \in [T]$, we establish the events $\mathcal{E}_t^{\boldsymbol{\theta}}$ and $\mathcal{E}_t^{\mathrm{TS}}$. These events are denoted as the inclusion of $\widehat{\boldsymbol{\theta}}_t$ and $\boldsymbol{\theta}_t^{\mathrm{TS}}$ in the pertinent confidence ellipse with some confidence radius $\widehat{\beta}_t$, respectively, *i.e.*,

$$\mathcal{E}_t^{\boldsymbol{\theta}} = \big\{ |\mathbf{x}^{\mathsf{T}}\widehat{\boldsymbol{\theta}}_t - \mathbf{x}^{\mathsf{T}}\boldsymbol{\theta}^*| \le \widehat{\beta}_t \|\mathbf{x}\|_{\mathbf{V}_t^{-1}}, \forall \mathbf{x} \in \mathcal{X}_t \big\}$$

and

$$\mathcal{E}_t^{\mathrm{TS}} = \big\{ |\mathbf{x}^{\mathsf{T}}\boldsymbol{\theta}_t^{\mathrm{TS}} - \mathbf{x}^{\mathsf{T}}\widehat{\boldsymbol{\theta}}_t| \le 2\widehat{\beta}_t \sqrt{d \log t} \|\mathbf{x}\|_{\mathbf{V}_t^{-1}}, \forall \mathbf{x} \in \mathcal{X}_t \big\}.$$

We proceed to define the set $\mathcal{S}(t)$ representing the saturated arms at round $t$ as follows:

$$\mathcal{S}(t) = \big\{ \mathbf{x} \in \mathcal{X}_t : \Delta_t(\mathbf{x}) > (2\sqrt{d \log t} + 1)\widehat{\beta}_t \|\mathbf{x}\|_{\mathbf{V}_t^{-1}} \big\}, \tag{A.1}$$

where we denote $\Delta_t(\mathbf{x}) = \langle \mathbf{x}_t^* - \mathbf{x}, \boldsymbol{\theta}^* \rangle$ as the suboptimality of $\mathbf{x} \in \mathcal{X}_t$. Moreover, we present a filtration, denoted by $\{\mathcal{H}_t\}_{t \ge 1}$, which is defined as

$$\mathcal{H}_t = \{\mathcal{X}_1, \boldsymbol{\theta}_1^{\mathrm{TS}}, x_1, \overline{\sigma}_1, r_1, \dots, \mathcal{X}_{t-1}, \boldsymbol{\theta}_{t-1}^{\mathrm{TS}}, x_{t-1}, \overline{\sigma}_{t-1}, r_{t-1}, \mathcal{X}_t\}.$$

This filtration encapsulates the information that has been gleaned up until the observation of the set of context vectors $\mathcal{X}_t$ at each particular round.

Concentration inequalities are widely used in literature (Fei and Xu, 2022b; Min et al., 2022b; He et al., 2022; Lu et al., 2022; Xu et al., 2023), and here we introduce a Bernstein-type inequality applicable to vector-valued martingales. This theorem, when coalesced with the approach of weighted ridge regression, enables us to construct stringent bounds that are dependent on the variance $\{\sigma_t^2\}_{t \ge 1}$.

**Theorem A.1** (Zhou and Gu (2022), Theorem 4.3). *Let $\{\mathcal{H}_t\}_{t=1}^{\infty}$ be a filtration, $\{\mathbf{y}_t, \varrho_t\}_{t \ge 1}$ a stochastic process so that for each $t \ge 1$, $\mathbf{y}_t \in \mathbb{R}^d$ is $\mathcal{H}_t$-measurable and $\varrho_t \in \mathbb{R}$ is $\mathcal{H}_{t+1}$-measurable. For each $t \ge 1$, define $r_t := \langle \mathbf{y}_t, \boldsymbol{\theta}^* \rangle + \varrho_t$ and suppose that $\varrho_t$, $\mathbf{y}_t$ satisfy*

$$\|\mathbf{y}_t\| \le L, \quad |\varrho_t| \le R, \quad \mathbb{E}[\varrho_t \mid \mathcal{H}_t] = 0, \quad \mathbb{E}[\varrho_t^2 \mid \mathcal{H}_t] \le \sigma^2$$

*for some constants $L, R, \sigma, \epsilon > 0$ and $\boldsymbol{\theta}^* \in \mathbb{R}^d$. Then, for any $0 < \delta < 1$, with probability at least $1 - \delta$ we have for all $t > 0$ that*

$$\Big\| \sum_{s=1}^{t} \varrho_s \mathbf{y}_s \Big\|_{\mathbf{U}_t^{-1}} \le \beta_t, \quad \|\boldsymbol{\theta}_t - \boldsymbol{\theta}^*\|_{\mathbf{U}_t} \le \beta_t + \sqrt{\lambda}\|\boldsymbol{\theta}^*\|,$$

*where $\mathbf{U}_t = \lambda \mathbf{I}_d + \sum_{s=1}^{t} \mathbf{y}_s \mathbf{y}_s^{\mathsf{T}}$, $\boldsymbol{\nu}_t = \sum_{s=1}^{t} r_s \mathbf{y}_s$, $\boldsymbol{\theta}_t = \mathbf{U}_t^{-1} \boldsymbol{\nu}_t$, and*

$$\beta_t = 12\sigma \sqrt{d \cdot \log\Big(1 + \frac{tL^2}{d\lambda}\Big) \log\Big(32\Big(1 + \log\frac{R}{\epsilon}\Big)\frac{t^2}{\delta}\Big)} + 6\log\Big(32\Big(1 + \log\frac{R}{\epsilon}\Big)\frac{t^2}{\delta}\Big)\epsilon$$

$$+ 24\log\Big(32\Big(1 + \log\frac{R}{\epsilon}\Big)\frac{t^2}{\delta}\Big) \max_{s \in [t]}\{|\varrho_s| \min\{1, \|\mathbf{y}_s\|_{\mathbf{U}_{s-1}^{-1}}\}\}.$$

## A.2  Proof of supporting lemmata

The forthcoming Lemmas A.2 and A.3 show that the events $\mathcal{E}_t^{\boldsymbol{\theta}}$ and $\mathcal{E}_t^{\mathrm{TS}}$ hold with high probability for all $t \in [T]$. The anti-concentration argument can then be developed in a manner analogous to the approach employed by Agrawal and Goyal (2013), which is often not necessary in UCB-type of arguments (Fei and Xu, 2022a; Chu et al., 2011; Min et al., 2023; Zhao et al., 2023).

**Lemma A.2.** *Under Assumption 3.1, for any $\delta \in (0, 1)$, if we define for all $t \ge 1$ that*

$$\widehat{\beta}_t = 12\sqrt{d \cdot \log\Big(1 + \frac{t}{d\lambda\alpha^2}\Big)\log\Big(64\Big(1 + \log\frac{\gamma^2}{\alpha}\Big)\frac{t^2}{\delta}\Big)}$$

$$+ \frac{30R}{\gamma^2} \cdot \log\Big(64\Big(1 + \log\frac{\gamma^2}{\alpha}\Big)\frac{t^2}{\delta}\Big) + \sqrt{\lambda} \tag{A.2}$$

with the coefficients $\alpha, \gamma, \lambda$ introduced in Algorithm 2, then the estimate $\widehat{\boldsymbol{\theta}}_t$ given by the associated weighted regressions satisfy

$$\mathbb{P}\left(|\mathbf{x}^\mathsf{T}\widehat{\boldsymbol{\theta}}_t - \mathbf{x}^\mathsf{T}\boldsymbol{\theta}^*| \leq \widehat{\beta}_t \|\mathbf{x}\|_{\mathbf{V}_t^{-1}}, \forall \mathbf{x} \in \mathfrak{X}_t, \forall t \geq 1\right) \geq 1 - \delta/2.$$

*Proof.* Observe that for any $t \geq 1$ and $\mathbf{x} \in \mathfrak{X}_t$, by Cauchy-Schwarz inequality we have

$$|\mathbf{x}^\mathsf{T}\widehat{\boldsymbol{\theta}}_t - \mathbf{x}^\mathsf{T}\boldsymbol{\theta}^*| \leq \|\widehat{\boldsymbol{\theta}}_t - \boldsymbol{\theta}^*\|_{\mathbf{V}_t} \|\mathbf{x}\|_{\mathbf{V}_t^{-1}}, \tag{A.3}$$

so it suffices to bound each $\|\mathbf{x}\|_{\mathbf{V}_t^{-1}}$. By Assumption 3.1, the following conditions on $\varepsilon_t$ and $\mathbf{x}_t$ hold:

$$\left|\frac{\varepsilon_t}{\overline{\sigma}_t}\right| \leq \frac{R}{\alpha}, \quad \mathbb{E}[\varepsilon_t \mid \mathscr{H}_t] = 0, \quad \mathbb{E}\left[\left(\frac{\varepsilon_t}{\overline{\sigma}_t}\right)^2 \;\middle|\; \mathscr{H}_t\right] \leq 1, \quad \left\|\frac{\mathbf{x}_t}{\overline{\sigma}_t}\right\| \leq \frac{1}{\alpha}.$$

Moreover, we can obtain following $\overline{\sigma}_t = \max\{\sigma_t, \alpha, \gamma\|\mathbf{x}_t\|_{\mathbf{V}_t^{-1}}^{1/2}\}$ that

$$\left|\frac{\varepsilon_t}{\overline{\sigma}_t}\right| \cdot \min\left\{1, \left\|\frac{\mathbf{x}_t}{\overline{\sigma}_t}\right\|_{\mathbf{V}_t^{-1}}\right\} \leq \frac{R}{\overline{\sigma}_t^2}\|\mathbf{x}_t\|_{\mathbf{V}_t^{-1}} \leq \frac{R}{\gamma^2}.$$

Therefore, by applying Theorem A.1 with the stochastic process $\{\mathbf{x}_t/\overline{\sigma}_t, \varepsilon_t/\overline{\sigma}_t\}_{t\geq 1}$ and $\epsilon = R/\gamma^2$, we deduce that with probability at least $1 - \delta/2$ for all $t \geq 1$

$$\|\widehat{\boldsymbol{\theta}}_t - \boldsymbol{\theta}^*\|_{\mathbf{V}_t} \leq 12\sqrt{d \cdot \log\left(1 + \frac{t}{d\lambda\alpha^2}\right)\log\left(64\left(1 + \log\frac{\gamma^2}{\alpha}\right)\frac{t^2}{\delta}\right)}$$
$$+ \frac{30R}{\gamma^2} \cdot \log\left(64\left(1 + \log\frac{\gamma^2}{\alpha}\right)\frac{t^2}{\delta}\right) + \sqrt{\lambda}\|\boldsymbol{\theta}^*\|.$$

Since $\|\boldsymbol{\theta}^*\|_2 \leq 1$ by Assumption 3.1, it follows from the definition of $\widehat{\beta}_t$ that $\|\widehat{\boldsymbol{\theta}}_t - \boldsymbol{\theta}^*\|_{\mathbf{V}_t} \leq \widehat{\beta}_t$. Combining this with (A.3), we conclude that with probability at least $1 - \delta/2$,

$$|\mathbf{x}^\mathsf{T}\widehat{\boldsymbol{\theta}}_t - \mathbf{x}^\mathsf{T}\boldsymbol{\theta}^*| \leq \widehat{\beta}_t \|\mathbf{x}\|_{\mathbf{V}_t^{-1}}$$

for all $t \geq 1$ and $\mathbf{x} \in \mathfrak{X}_t$. $\qquad\square$

The following lemma establishes the control over $|\mathbf{x}^\mathsf{T}\boldsymbol{\theta}_t^{\mathrm{TS}} - \mathbf{x}^\mathsf{T}\widehat{\boldsymbol{\theta}}_t|$.

**Lemma A.3.** *Under Assumption 3.1, for any $\delta \in (0,1)$, let $\widehat{\beta}_t$ be as defined in Lemma A.2 for all $t \geq 1$. Then the sampled variable $\boldsymbol{\theta}_t^{\mathrm{TS}}$ in Algorithm 2 satisfies*

$$\mathbb{P}\left(|\mathbf{x}^\mathsf{T}\boldsymbol{\theta}_t^{\mathrm{TS}} - \mathbf{x}^\mathsf{T}\widehat{\boldsymbol{\theta}}_t| \leq \widehat{\beta}_t\sqrt{2d\log(4t^2/\delta)} \cdot \|\mathbf{x}\|_{\mathbf{V}_t^{-1}}, \forall \mathbf{x} \in \mathfrak{X}_t, \forall t \geq 1\right) \geq 1 - \frac{\delta}{2}.$$

*Proof.* By definition, $\boldsymbol{\theta}_t^{\mathrm{TS}} \sim \mathcal{N}(\widehat{\boldsymbol{\theta}}_t, \boldsymbol{\Sigma}_t)$ where $\boldsymbol{\Sigma}_t = \widehat{\beta}_t^2 \mathbf{V}_t^{-1}$, thus $\|\boldsymbol{\theta}_t^{\mathrm{TS}} - \widehat{\boldsymbol{\theta}}_t\|_{\boldsymbol{\Sigma}_t^{-1}} \sim \mathcal{N}(0,1)$. Then using the concentration inequality in Lemma C.1 for standard normal distribution, for every $t \in [T]$, we have with probability at least $1 - \frac{\delta}{4t^2}$ that

$$|\mathbf{x}^\mathsf{T}\boldsymbol{\theta}_t^{\mathrm{TS}} - \mathbf{x}^\mathsf{T}\widehat{\boldsymbol{\theta}}_t| = |\mathbf{x}^\mathsf{T}\mathbf{V}_t^{-1/2}\mathbf{V}_t^{1/2}(\boldsymbol{\theta}_t^{\mathrm{TS}} - \widehat{\boldsymbol{\theta}}_t)|$$
$$\leq \widehat{\beta}_t \left\|\frac{\mathbf{V}_t^{1/2}(\boldsymbol{\theta}_t^{\mathrm{TS}} - \widehat{\boldsymbol{\theta}}_t)}{\widehat{\beta}_t}\right\|_2 \|\mathbf{x}\|_{\mathbf{V}_t^{-1}}$$
$$\leq \widehat{\beta}_t\sqrt{2d\log(4t^2/\delta)}\|\mathbf{x}\|_{\mathbf{V}_t^{-1}},$$

where the first inequality follows from Cauchy-Schwarz inequality. Applying a union bound over $t \in [T]$ and utilizing the fact that $\sum_{t=1}^\infty \frac{1}{t^2} \leq 2$, we obtain the final result. $\qquad\square$

In the following, we delve into the anti-concentration argument presented in Lemmas A.4 to A.6, which constitute the upper bound on expected regret.

**Lemma A.4.** *For any $t \geq 1$, if $\mathcal{H}_t$ is a filtration such that $\mathcal{E}_t^{\boldsymbol{\theta}}$ holds, we have a constant probability that the reward estimate $\langle \mathbf{x}_t^*, \boldsymbol{\theta}_t^{\mathrm{TS}} \rangle$ of the optimal arm constitutes an upper confidence bound, i.e.,*

$$\mathbb{P}(\mathbf{x}_t^{*\mathsf{T}}\boldsymbol{\theta}_t^{\mathrm{TS}} > \mathbf{x}_t^{*\mathsf{T}}\boldsymbol{\theta}^* \mid \mathcal{H}_t) \geq \frac{1}{4e\sqrt{\pi}}.$$

*Proof.* Note that we can bound the quantity $J_t = (\mathbf{x}_t^{*\mathsf{T}}\boldsymbol{\theta}^* - \mathbf{x}_t^{*\mathsf{T}}\widehat{\boldsymbol{\theta}}_t)/(\widehat{\beta}_t \|\mathbf{x}_t^*\|_{\mathbf{V}_t^{-1}})$ as follows:

$$
\begin{aligned}
|J_t| &= \left| \frac{\mathbf{x}_t^{*\mathsf{T}}\boldsymbol{\theta}^* - \mathbf{x}_t^{*\mathsf{T}}\widehat{\boldsymbol{\theta}}_t}{\widehat{\beta}_t \|\mathbf{x}_t^*\|_{\mathbf{V}_t^{-1}}} \right| \\
&\leq \frac{\widehat{\beta}_t \|\mathbf{x}_t^*\|_{\mathbf{V}_t^{-1}}}{\widehat{\beta}_t \|\mathbf{x}_t^*\|_{\mathbf{V}_t^{-1}}} \\
&= 1,
\end{aligned}
\tag{A.4}
$$

where the inequality is due to Lemma A.2. Recalling that $\boldsymbol{\theta}_t^{\mathrm{TS}}$ has a mean of $\widehat{\boldsymbol{\theta}}_t$, we can establish that the probability of the reward estimate $\mathbf{x}_t^{*\mathsf{T}}\boldsymbol{\theta}_t^{\mathrm{TS}}$ serving as an upper bound for the true reward is lower-bounded by

$$
\begin{aligned}
\mathbb{P}(\mathbf{x}_t^{*\mathsf{T}}\boldsymbol{\theta}_t^{\mathrm{TS}} > \mathbf{x}_t^{*\mathsf{T}}\boldsymbol{\theta}^* \mid \mathcal{H}_t) &= \mathbb{P}(\mathbf{x}_t^{*\mathsf{T}}\boldsymbol{\theta}_t^{\mathrm{TS}} - \mathbf{x}_t^{*\mathsf{T}}\widehat{\boldsymbol{\theta}}_t > \mathbf{x}_t^{*\mathsf{T}}\boldsymbol{\theta}^* - \mathbf{x}_t^{*\mathsf{T}}\widehat{\boldsymbol{\theta}}_t \mid \mathcal{H}_t) \\
&= \mathbb{P}\left( \frac{\mathbf{x}_t^{*\mathsf{T}}\boldsymbol{\theta}_t^{\mathrm{TS}} - \mathbf{x}_t^{*\mathsf{T}}\widehat{\boldsymbol{\theta}}_t}{\widehat{\beta}_t \|\mathbf{x}_t^*\|_{\mathbf{V}_t^{-1}}} > \frac{\mathbf{x}_t^{*\mathsf{T}}\boldsymbol{\theta}^* - \mathbf{x}_t^{*\mathsf{T}}\widehat{\boldsymbol{\theta}}_t}{\widehat{\beta}_t \|\mathbf{x}_t^*\|_{\mathbf{V}_t^{-1}}} \,\middle|\, \mathcal{H}_t \right) \\
&\geq \frac{1}{4\sqrt{\pi}} e^{-J_t^2}.
\end{aligned}
\tag{A.5}
$$

The last inequality follows from the anti-concentration inequality in Lemma C.1 for standard normal distribution and the fact that

$$\frac{\mathbf{x}_t^{*\mathsf{T}}\boldsymbol{\theta}_t^{\mathrm{TS}} - \mathbf{x}_t^{*\mathsf{T}}\widehat{\boldsymbol{\theta}}_t}{\widehat{\beta}_t \|\mathbf{x}_t^*\|_{\mathbf{V}_t^{-1}}} \sim \mathcal{N}(0, 1).$$

Consequently, combining (A.4) and (A.5), it follows that the desired probability is lower bounded by a constant, *i.e.*, $\mathbb{P}(\mathbf{x}_t^{*\mathsf{T}}\boldsymbol{\theta}_t^{\mathrm{TS}} > \mathbf{x}_t^{*\mathsf{T}}\boldsymbol{\theta}^* \mid \mathcal{H}_t) \geq \frac{1}{4e\sqrt{\pi}}$. $\qquad\square$

**Lemma A.5.** *For any $t \geq 1$, if $\mathcal{H}_t$ is a filtration such that $\mathcal{E}_t^{\boldsymbol{\theta}}$ holds, we have with a constant probability that the chosen arm $\mathbf{x}_t$ is not a saturated arm, i.e.,*

$$\mathbb{P}(\mathbf{x}_t \notin \mathcal{S}(t) \mid \mathcal{H}_t) \geq \frac{1}{4e\sqrt{\pi}} - \frac{1}{t^2}.$$

*Proof.* Recall the definition of the set of saturated arms $\mathcal{S}(t)$ in (A.1) and that the selected arm $\mathbf{x}_t = \arg\max_{\mathbf{x} \in \mathcal{X}_t} \mathbf{x}^\mathsf{T}\boldsymbol{\theta}_t^{\mathrm{TS}}$ aims to maximize the estimated reward. Consequently, $\mathbf{x}_t \notin \mathcal{S}(t)$ if $\mathbf{x}_t^{*\mathsf{T}}\boldsymbol{\theta}_t^{\mathrm{TS}} \geq \mathbf{x}^\mathsf{T}\boldsymbol{\theta}_t^{\mathrm{TS}}$ for all saturated arms $\mathbf{x} \in \mathcal{S}(t)$, implying that the estimated reward for the optimal arm surpasses the estimated rewards of all saturated arms. It follows that:

$$\mathbb{P}(\mathbf{x}_t \notin \mathcal{S}(t) \mid \mathcal{H}_t) \geq \mathbb{P}(\mathbf{x}_t^{*\mathsf{T}}\boldsymbol{\theta}_t^{\mathrm{TS}} \geq \mathbf{x}^\mathsf{T}\boldsymbol{\theta}_t^{\mathrm{TS}}, \forall \mathbf{x} \in \mathcal{S}(t) \mid \mathcal{H}_t).$$

Further note that by the definition of saturated arms, we have

$$\mathbf{x}^\mathsf{T}\boldsymbol{\theta}_t^{\mathrm{TS}} \leq \mathbf{x}^\mathsf{T}\boldsymbol{\theta}^* + (2\sqrt{d\log t} + 1)\widehat{\beta}_t \|\mathbf{x}\|_{\mathbf{V}_t^{-1}} < \mathbf{x}^\mathsf{T}\boldsymbol{\theta}^* + \Delta_t(\mathbf{x}) = \mathbf{x}_t^{*\mathsf{T}}\boldsymbol{\theta}^*$$

when both $\mathcal{E}_t^{\boldsymbol{\theta}}$ and $\mathcal{E}_t^{\mathrm{TS}}$ hold. Therefore,

$$\mathbb{P}(\mathbf{x}_t \notin \mathcal{S}(t) \mid \mathcal{H}_t) \geq \mathbb{P}(\mathbf{x}_t^{*\mathsf{T}}\boldsymbol{\theta}_t^{\mathrm{TS}} > \mathbf{x}_t^{*\mathsf{T}}\boldsymbol{\theta}^* \mid \mathcal{H}_t) - \mathbb{P}(\overline{\mathcal{E}^{\mathrm{TS}}}(t) \mid \mathcal{H}_t),$$

The last inequality is a consequence of Lemmas A.3 and A.4. $\qquad\square$

**Lemma A.6.** *For any $t \geq 1$, if $\mathscr{H}_t$ is a filtration such that both $\mathcal{E}_t^{\boldsymbol{\theta}}$ and $\mathcal{E}_t^{\mathrm{TS}}$ hold, we have*

$$\mathbb{E}[\Delta_t \mid \mathscr{H}_t] \leq \left( \frac{2}{\frac{1}{4e\sqrt{\pi}} - \frac{1}{t^2}} + 1 \right) (2\sqrt{d \log t} + 1)\widehat{\beta}_t \, \mathbb{E}[\|\mathbf{x}_t\|_{\mathbf{V}_t^{-1}} \mid \mathscr{H}_t] + \frac{2}{t^2}.$$

*Proof.* Let $\mathbf{x}_t^{\dagger}$ represent the unsaturated arm with the smallest $\|\cdot\|_{\mathbf{V}_t^{-1}}$ norm, *i.e.*,

$$\mathbf{x}_t^{\dagger} = \operatorname*{arg\,min}_{\mathbf{x} \notin \mathcal{S}(t)} \|\mathbf{x}\|_{\mathbf{V}_t^{-1}}.$$

The existence of such an unsaturated arm $\mathbf{x}_t^{\dagger}$ is guaranteed since $\mathbf{x}_t^* \notin \mathcal{S}(t)$. When both $\mathcal{E}_t^{\boldsymbol{\theta}}$ and $\mathcal{E}_t^{\mathrm{TS}}$ hold, we can express the suboptimality as follows:

$$\begin{aligned}
\Delta_t &= \Delta_t(\mathbf{x}_t^{\dagger}) + {\mathbf{x}_t^{\dagger}}^{\mathsf{T}} \boldsymbol{\theta}^* - \mathbf{x}_t^{\mathsf{T}} \boldsymbol{\theta}^* \\
&\leq \Delta_t(\mathbf{x}_t^{\dagger}) + ({\mathbf{x}_t^{\dagger}}^{\mathsf{T}} \boldsymbol{\theta}_t^{\mathrm{TS}} + (2\sqrt{d \log t} + 1)\widehat{\beta}_t \|\mathbf{x}_t^{\dagger}\|_{\mathbf{V}_t^{-1}}) \\
&\quad - (\mathbf{x}_t^{\mathsf{T}} \boldsymbol{\theta}_t^{\mathrm{TS}} - (2\sqrt{d \log t} + 1)\widehat{\beta}_t \|\mathbf{x}_t\|_{\mathbf{V}_t^{-1}})
\end{aligned}$$

Then since $\mathbf{x}_t$ is the optimal arm under $\boldsymbol{\theta}_t^{\mathrm{TS}}$, we further have

$$\begin{aligned}
\Delta_t &\leq \Delta_t(\mathbf{x}_t^{\dagger}) + (2\sqrt{d \log t} + 1)\widehat{\beta}_t \|\mathbf{x}_t^{\dagger}\|_{\mathbf{V}_t^{-1}} + (2\sqrt{d \log t} + 1)\widehat{\beta}_t \|\mathbf{x}_t\|_{\mathbf{V}_t^{-1}} \\
&\leq (2\sqrt{d \log t} + 1)\widehat{\beta}_t (2\|\mathbf{x}_t^{\dagger}\|_{\mathbf{V}_t^{-1}} + \|\mathbf{x}_t\|_{\mathbf{V}_t^{-1}}),
\end{aligned}$$

where the second inequality results from $\mathbf{x}_t^{\dagger} \notin \mathcal{S}(t)$. Note that $\|\mathbf{x}_t\|_{\mathbf{V}_t^{-1}} \geq \|\mathbf{x}_t^{\dagger}\|_{\mathbf{V}_t^{-1}}$ with constant probability, *i.e.*,

$$\begin{aligned}
\mathbb{E}[\|\mathbf{x}_t\|_{\mathbf{V}_t^{-1}} \mid \mathscr{H}_t] &\geq \mathbb{E}[\|\mathbf{x}_t\|_{\mathbf{V}_t^{-1}} \mid \mathscr{H}_t, \mathbf{x}_t \notin \mathcal{S}(t)] \cdot \mathbb{P}(\mathbf{x}_t \notin \mathcal{S}(t) \mid \mathscr{H}_t) \\
&\geq \left( \frac{1}{4e\sqrt{\pi}} - \frac{1}{t^2} \right) \mathbb{E}[\|\mathbf{x}_t^{\dagger}\|_{\mathbf{V}_t^{-1}} \mid \mathscr{H}_t].
\end{aligned} \tag{A.6}$$

The last inequality is due to Lemma A.5 and the definition of $\mathbf{x}_t^{\dagger}$ as the unsaturated arm with the smallest $\|\cdot\|_{\mathbf{V}_t^{-1}}$. Consequently,

$$\begin{aligned}
\mathbb{E}[\Delta_t \mid \mathscr{H}_t] &\leq \mathbb{E}[(2\sqrt{d \log t} + 1)\widehat{\beta}_t (2\|\mathbf{x}_t^{\dagger}\|_{\mathbf{V}_t^{-1}} + \|\mathbf{x}_t\|_{\mathbf{V}_t^{-1}}) \mid \mathscr{H}_t] + 2\,\mathbb{P}(\overline{\mathcal{E}_t^{\mathrm{TS}}}) \\
&\leq \left( \frac{2}{\frac{1}{4e\sqrt{\pi}} - \frac{1}{t^2}} + 1 \right) (2\sqrt{d \log t} + 1)\widehat{\beta}_t \, \mathbb{E}[\|\mathbf{x}_t\|_{\mathbf{V}_t^{-1}} \mid \mathscr{H}_t] + \frac{2}{t^2},
\end{aligned}$$

where the first inequality is due to $\Delta_t \leq 2$ for all $t$, the second inequality follows from applying (A.6), Lemmas A.3 and A.4. $\square$

### A.3 Proof of Theorem 4.1

Let us define a shorthand $\mathcal{E}^{\boldsymbol{\theta}} = \cap_{t=1}^T \mathcal{E}_t^{\boldsymbol{\theta}}$. The expected regret can be decomposed as follows:

$$\begin{aligned}
\mathbb{E}[\mathcal{R}(T)] &= \sum_{t=1}^T \mathbb{E}[\Delta_t \mid \mathcal{E}^{\boldsymbol{\theta}}]\,\mathbb{P}(\mathcal{E}^{\boldsymbol{\theta}}) + \sum_{t=1}^T \mathbb{E}[\Delta_t \mid \overline{\mathcal{E}^{\boldsymbol{\theta}}}]\,\mathbb{P}(\overline{\mathcal{E}^{\boldsymbol{\theta}}}) \\
&\leq \sum_{t=1}^T \mathbb{E}[\Delta_t \mid \mathscr{H}_t, \mathcal{E}^{\boldsymbol{\theta}}] + 2T \cdot \frac{\delta}{2} \\
&\leq \sum_{t=1}^T \mathbb{E}[\min\{2, (2\sqrt{d \log t} + 1)C\widehat{\beta}_t \|\mathbf{x}_t\|_{\mathbf{V}_t^{-1}}\} \mid \mathscr{H}_t] + \sum_{t=1}^T \frac{2}{t^2} + \delta T \\
&= \mathbb{E}\left[ \sum_{t=1}^T \min\{2, (2\sqrt{d \log t} + 1)C\widehat{\beta}_t \|\mathbf{x}_t\|_{\mathbf{V}_t^{-1}}\} \,\middle|\, \mathcal{E}^{\boldsymbol{\theta}} \right] + \delta T + 4,
\end{aligned}$$

where we write $C = \max_{t \geq 1} \frac{2}{\left|\frac{1}{4e\sqrt{\pi}} - \frac{1}{t^2}\right|} + 1$. The first inequality is due to Lemma A.2, the second inequality is due to Lemma A.6, and the last inequality follows from $\Delta_t \leq 2$ for all $t \in [T]$. To alleviate the burden on notation, we write $g_t = \min\{2, (2\sqrt{d\log t} + 1)C\widehat{\beta}_t \|\mathbf{x}_t\|_{\mathbf{V}_t^{-1}}\}$ and an index set $\mathcal{T} = \{t \in [T] : \|\mathbf{x}_t/\overline{\sigma}_t\|_{\mathbf{V}_t^{-1}} \geq 1\}$. It follows that

$$
\begin{aligned}
\sum_{t=1}^{T} g_t &= \sum_{t=1}^{T} \min\{2, (2\sqrt{d\log t} + 1)C\widehat{\beta}_t \overline{\sigma}_t \|\mathbf{x}_t/\overline{\sigma}_t\|_{\mathbf{V}_t^{-1}}\} \\
&= \sum_{t \in \mathcal{T}} \min\{2, (2\sqrt{d\log t} + 1)C\widehat{\beta}_t \overline{\sigma}_t \|\mathbf{x}_t/\overline{\sigma}_t\|_{\mathbf{V}_t^{-1}}\} \\
&\qquad + \sum_{t \notin \mathcal{T}} \min\{2, (2\sqrt{d\log t} + 1)C\widehat{\beta}_t \overline{\sigma}_t \|\mathbf{x}_t/\overline{\sigma}_t\|_{\mathbf{V}_t^{-1}}\} \\
&\leq 2|\mathcal{T}| + \sum_{t \notin \mathcal{T}} (2\sqrt{d\log t} + 1)C\widehat{\beta}_t \overline{\sigma}_t \|\mathbf{x}_t/\overline{\sigma}_t\|_{\mathbf{V}_t^{-1}} \\
&= 2|\mathcal{T}| + 3C \sum_{t \notin \mathcal{T}} \widehat{\beta}_t \sqrt{d\log t}\,\overline{\sigma}_t \min\{1, \|\mathbf{x}_t/\overline{\sigma}_t\|_{\mathbf{V}_t^{-1}}\} \\
&\leq 4d\log(1 + T/(d\lambda\alpha^2)) + 3C \sum_{t \notin \mathcal{T}} \widehat{\beta}_t \sqrt{d\log t}\,\overline{\sigma}_t \min\{1, \|\mathbf{x}_t/\overline{\sigma}_t\|_{\mathbf{V}_t^{-1}}\}, \qquad \text{(A.7)}
\end{aligned}
$$

where the first inequality follows from the definition of $g_t$, the last equality is due to $\|\mathbf{x}_t/\overline{\sigma}_t\|_{\mathbf{V}_t^{-1}} < 1$ for all $t \notin \mathcal{T}$, and the second inequality is due to

$$
\begin{aligned}
|\mathcal{T}| &\leq \sum_{t \in \mathcal{T}} \min\{1, \|\mathbf{x}_t/\overline{\sigma}_t\|_{\mathbf{V}_t^{-1}}^2\} \\
&\leq \sum_{t=1}^{T} \min\{1, \|\mathbf{x}_t/\overline{\sigma}_t\|_{\mathbf{V}_t^{-1}}^2\} \\
&\leq 2d\log(1 + T/(d\lambda\alpha^2)).
\end{aligned}
$$

The last inequality is due to Lemma C.2 and $\|\mathbf{x}_t/\overline{\sigma}_t\| \leq 1/\alpha$. Further, we decompose $\overline{\mathcal{T}} = [T]\backslash\mathcal{T} = \overline{\mathcal{T}}_1 \cup \overline{\mathcal{T}}_2$ where

$$
\overline{\mathcal{T}}_1 = \{t \in \overline{\mathcal{T}} : \overline{\sigma}_t = \sigma_t \text{ or } \overline{\sigma}_t = \alpha\}, \qquad \overline{\mathcal{T}}_1 = \{t \in \overline{\mathcal{T}} : \overline{\sigma}_t = \gamma\|\mathbf{x}_t\|_{\mathbf{V}_t^{-1}}^{1/2}\}.
$$

The second term in (A.7) can be controlled through a decomposition

$$
\begin{aligned}
\sum_{t \notin \mathcal{T}} \widehat{\beta}_t \sqrt{d\log t}\,\overline{\sigma}_t \min\{1, \|\mathbf{x}_t/\overline{\sigma}_t\|_{\mathbf{V}_t^{-1}}\} &= \sum_{t \in \overline{\mathcal{T}}_1} \widehat{\beta}_t \sqrt{d\log t}\,\overline{\sigma}_t \min\{1, \|\mathbf{x}_t/\overline{\sigma}_t\|_{\mathbf{V}_t^{-1}}\} \\
&\qquad + \sum_{t \in \overline{\mathcal{T}}_2} \widehat{\beta}_t \sqrt{d\log t}\,\overline{\sigma}_t \min\{1, \|\mathbf{x}_t/\overline{\sigma}_t\|_{\mathbf{V}_t^{-1}}\},
\end{aligned}
$$

For $t \in \overline{\mathcal{T}}_1$, we have

$$
\begin{aligned}
\sum_{t \in \overline{\mathcal{T}}_1} \widehat{\beta}_t \sqrt{d\log t}\,\overline{\sigma}_t \min\{1, \|\mathbf{x}_t/\overline{\sigma}_t\|_{\mathbf{V}_t^{-1}}\} &\leq \sum_{t \in \overline{\mathcal{T}}_1} \widehat{\beta}_t \sqrt{d\log t}(\sigma_t + \alpha) \min\{1, \|\mathbf{x}_t/\overline{\sigma}_t\|_{\mathbf{V}_t^{-1}}\} \\
&\leq \sum_{t=1}^{T} \widehat{\beta}_t \sqrt{d\log t}(\sigma_t + \alpha) \min\{1, \|\mathbf{x}_t/\overline{\sigma}_t\|_{\mathbf{V}_t^{-1}}\} \\
&\leq \sqrt{2d\sum_{t=1}^{T} \widehat{\beta}_t^2 (\sigma_t^2 + \alpha^2)\log T} \sqrt{\sum_{t=1}^{T} \min\{1, \|\mathbf{x}_t/\overline{\sigma}_t\|_{\mathbf{V}_t^{-1}}^2\}} \\
&\leq 2\sqrt{d\sum_{t=1}^{T} \widehat{\beta}_t^2 (\sigma_t^2 + \alpha^2)\log T} \sqrt{d\log(1 + T/(d\lambda\alpha^2))},
\end{aligned}
$$

where the first inequality is due to Cauchy-Schwarz inequality, and the second inequality is due to Lemma C.2 and $\|\mathbf{x}_t/\overline{\sigma}_t\| \le 1/\alpha$. For $t \in \overline{\mathcal{T}}_2$, we have $\overline{\sigma}_t = \gamma^2 \|\mathbf{x}_t/\overline{\sigma}_t\|_{\mathbf{V}_t^{-1}}$, and then

$$
\begin{aligned}
\sum_{t \in \overline{\mathcal{T}}_2} \widehat{\beta}_t \sqrt{d \log t} \overline{\sigma}_t \min\{1, \|\mathbf{x}_t/\overline{\sigma}_t\|_{\mathbf{V}_t^{-1}}\} &= \sum_{t \in \overline{\mathcal{T}}_2} \widehat{\beta}_t \sqrt{d \log t} \overline{\sigma}_t \|\mathbf{x}_t/\overline{\sigma}_t\|_{\mathbf{V}_t^{-1}} \\
&\le \gamma^2 \sum_{t \in \overline{\mathcal{T}}_2} \widehat{\beta}_t \sqrt{d \log t} \|\mathbf{x}_t/\overline{\sigma}_t\|_{\mathbf{V}_t^{-1}}^2 \\
&\le \gamma^2 \sum_{t=1}^{T} \widehat{\beta}_t \sqrt{d \log t} \min\{1, \|\mathbf{x}_t/\overline{\sigma}_t\|_{\mathbf{V}_t^{-1}}^2\} \\
&\le 2\gamma^2 d \sqrt{d \log T} \log(1 + T/(d\lambda\alpha^2))(\max_{t \in [T]} \widehat{\beta}_t).
\end{aligned}
$$

Hence, combining the inequalities above, we have

$$
\begin{aligned}
\sum_{t=1}^{T} g_t \le{}& 4d \log(1 + T/(d\lambda\alpha^2)) + 3C \cdot 2 \sqrt{d \sum_{t=1}^{T} \widehat{\beta}_t^2 (\sigma_t^2 + \alpha^2) \log T} \sqrt{d \log(1 + T/(d\lambda\alpha^2))} \\
&+ 3C \cdot 2\gamma^2 d \sqrt{d \log T} \log(1 + T/(d\lambda\alpha^2))(\max_{t \in [T]} \widehat{\beta}_t),
\end{aligned}
$$

and an upper bound on expected regret follows noting that $\sum_{t=1}^{T} \min\{2, (2\sqrt{d \log t} + 1)C\widehat{\beta}_t \|\mathbf{x}_t\|_{\mathbf{V}_t^{-1}}\}$ enjoys a deterministic upper bound:

$$
\begin{aligned}
\mathbb{E}[\mathcal{R}(T)] \le{}& \sum_{t=1}^{T} g_t + \delta T + 4 \\
\le{}& 4d \log(1 + T/(d\lambda\alpha^2)) + 3C \cdot 2 \sqrt{d \sum_{t=1}^{T} \widehat{\beta}_t^2 (\sigma_t^2 + \alpha^2) \log T} \sqrt{d \log(1 + T/(d\lambda\alpha^2))} \\
&+ 3C \cdot 2\gamma^2 d \sqrt{d \log T} \log(1 + T/(d\lambda\alpha^2))(\max_{t \in [T]} \widehat{\beta}_t) + \delta T + 4
\end{aligned}
$$

Take $\lambda = d$, $\alpha = 1/\sqrt{T}$, and $\gamma = \sqrt{R}/d^{1/4}$, it follows that

$$
\log(1 + T/(d\lambda\alpha^2)) \le \log(T^2/d^2 + 1)
$$

and

$$
\begin{aligned}
\widehat{\beta}_t ={}& 12\sqrt{d \log(1 + t/(d\lambda\alpha^2)) \log(32(1 + \log(\gamma^2/\alpha))t^2/\delta)} \\
&+ 30R \log(32(1 + \log(\gamma^2/\alpha))t^2/\delta)/\gamma^2 + \sqrt{\lambda} \\
\le{}& 12\sqrt{d \log(1 + tT/d^2) \log(32(1 + \log(R\sqrt{T}/\sqrt{d}))t^2/\delta)} \\
&+ 30\sqrt{d} \log(32(1 + \log(R\sqrt{T}/\sqrt{d}))t^2/\delta) + \sqrt{d}.
\end{aligned}
$$

Combining Lemma A.2 and the inequalities above with $\delta = \frac{1}{T}$, we conclude that

$$
\begin{aligned}
\mathbb{E}[\mathcal{R}(T)] \lesssim{}& d \log T + d \log T \sqrt{\sum_{t=1}^{T} \widehat{\beta}_t^2 (\sigma_t^2 + \alpha^2)} + \gamma^2 \sqrt{d^3 \log^3 T}(\max_{t \in [T]} \widehat{\beta}_t) \\
\lesssim{}& d \log T + d \log T \sqrt{\sum_{t=1}^{T} \widehat{\beta}_t^2 (\sigma_t^2 + \alpha^2)} + d \log T \sqrt{d \log T} \\
\lesssim{}& d \log T \sqrt{\sum_{t=1}^{T} \widehat{\beta}_t^2 (\sigma_t^2 + 1/T)} + d \log T \sqrt{d \log T} \\
\lesssim{}& d^{3/2} \log T \sqrt{\sum_{t=1}^{T} \sigma_t^2} + d^{3/2} \log T.
\end{aligned}
$$

Recall that we have

$$\mathbb{E}[\mathcal{R}(T)] \leq \sum_{t=1}^{T} \mathbb{E}[\min\{2, (2\sqrt{d\log t}+1)C\widehat{\beta}_t\|\mathbf{x}_t\|_{\mathbf{V}_t^{-1}}\} \mid \mathscr{H}_t] + \sum_{t=1}^{T} \frac{2}{t^2},$$

For notational convenience, we further define $S_t = \sum_{s=1}^{t} D_t$ where $D_t = \Delta_t - g_t - \frac{2}{t^2}$, and

$$g_t = \min\{2, (2\sqrt{d\log t}+1)C\widehat{\beta}_t\|\mathbf{x}_t\|_{\mathbf{V}_t^{-1}}\}.$$

It follows that $\mathbb{E}[D_t \mid \mathscr{H}_t] \leq 0$ for all $t \in [T]$, and $\{S_t\}_{t\geq 1}$ therefore forms a super-martingale process with respect to the filtrations $\{\mathscr{H}_t\}_t$. Note that $|D_t| \leq 2g_t + 2 \leq 4$, and with Azuma-Hoeffding inequality we have with probability at least $1 - \delta/2$ that

$$\sum_{t=1}^{T} G_t \leq \sum_{t=1}^{T} g_t + \sum_{t=1}^{T} \frac{2}{t^2} + \sqrt{2\sum_{t=1}^{T} 4\log(2/\delta)}$$

$$\leq \sum_{t=1}^{T} \min\{2, (2\sqrt{d\log t}+1)C\widehat{\beta}_t\|\mathbf{x}_t\|_{\mathbf{V}_t^{-1}}\} + \sum_{t=1}^{T} \frac{2}{t^2} + 2\sqrt{2T\log(2/\delta)}$$

$$= \sum_{t=1}^{T} \min\{2, (2\sqrt{d\log t}+1)C\widehat{\beta}_t\overline{\sigma}_t\|\mathbf{x}_t/\overline{\sigma}_t\|_{\mathbf{V}_t^{-1}}\} + \sum_{t=1}^{T} \frac{2}{t^2} + 2\sqrt{2T\log(2/\delta)}$$

$$\lesssim d^{3/2}\log T \sqrt{\sum_{t=1}^{T} \sigma_t^2} + d^{3/2}\log T + \sqrt{T}.$$

## B Regret Analysis of Algorithm 3

### B.1 Clarification of notation

Analogous to Appendix A, we first introduce two concentration events $\mathcal{E}_{t,\ell}^{\boldsymbol{\theta}}$ and $\mathcal{E}_{t,\ell}^{\text{TS}}$ for each $t \in [T]$ and $\ell \in [L]$, *i.e.*,

$$\mathcal{E}_{t,\ell}^{\boldsymbol{\theta}} = \{|\mathbf{x}^{\mathsf{T}}\widehat{\boldsymbol{\theta}}_{t,\ell} - \mathbf{x}^{\mathsf{T}}\boldsymbol{\theta}^*| \leq \widetilde{\beta}_{t,\ell}\|\mathbf{x}\|_{\mathbf{V}_{t,\ell}^{-1}}, \forall \mathbf{x} \in \mathcal{X}_t\}$$

and

$$\mathcal{E}_{t,\ell}^{\text{TS}} = \{|\mathbf{x}^{\mathsf{T}}\boldsymbol{\theta}_{t,\ell}^{\text{TS}} - \mathbf{x}^{\mathsf{T}}\widehat{\boldsymbol{\theta}}_{t,\ell}| \leq \sqrt{2d\log(4t^2L/\delta)}\widehat{\beta}_{t,\ell}\|\mathbf{x}\|_{\mathbf{V}_{t,\ell}^{-1}}, \forall \mathbf{x} \in \mathcal{X}_t\}.$$

These events are also characterized as the inclusion of $\widehat{\boldsymbol{\theta}}_{t,\ell}$ and $\boldsymbol{\theta}_{t,\ell}^{\text{TS}}$ in the corresponding confidence ellipse, respectively. It is noteworthy that the event $\mathcal{E}_{t,\ell}^{\boldsymbol{\theta}}$ is defined using

$$\widetilde{\beta}_{t,\ell} = 16 \cdot 2^{-\ell}\sqrt{\sum_{s \in \Psi_{t,\ell}} w_s^2\sigma_s^2\log(8t^2L/\delta)} + 6 \cdot 2^{-\ell}R\log(8t^2L/\delta) + 2^{-\ell+1} \tag{B.1}$$

rather than the confidence radius $\widehat{\beta}_{t,\ell}$.

Subsequently, we introduce a Freedman-type concentration inequality pertinent to vector-valued martingales. This theorem, distinct from Theorem A.1, necessitates a nuanced control on the uncertainty term $\|\mathbf{y}_t\|_{\mathbf{U}_{t-1}^{-1}}$ for all $t \geq 1$. Consequently, it furnishes an even more stringent bound in comparison to Theorem A.1.

**Theorem B.1** (Zhao et al. (2023), Theorem 2.1). *Let $\{\mathscr{H}_t\}_{t=1}^{\infty}$ be a filtration, $\{\mathbf{y}_t, \varrho_t\}_{t\geq 1}$ a stochastic process so that $\mathbf{y}_t \in \mathbb{R}^d$ is $\mathscr{H}_t$-measurable and $\varrho_t \in \mathbb{R}$ is $\mathscr{H}_{t+1}$-measurable. For $t \geq 1$ let $r_t = \langle \mathbf{y}_t, \boldsymbol{\theta}^* \rangle + \varrho_t$ and suppose that $\varrho_t, \mathbf{y}_t$ satisfy*

$$|\varrho_t| \leq R, \quad \mathbb{E}[\varrho_t \mid \mathscr{H}_t] = 0, \quad \sum_{s=1}^{t} \mathbb{E}[\varrho_s^2 \mid \mathscr{H}_s] \leq v_t$$

*for some constants $R, \{v_t\}_t > 0$ and $\boldsymbol{\theta}^* \in \mathbb{R}^d$. Then, for any $0 < \delta < 1$, with probability at least $1 - \delta$ we have for all $t > 0$ that*

$$\Big\| \sum_{s=1}^t \varrho_s \mathbf{y}_s \Big\|_{\mathbf{U}_t^{-1}} \leq \beta_t, \quad \|\boldsymbol{\theta}_t - \boldsymbol{\theta}^*\|_{\mathbf{U}_t} \leq \beta_t + \sqrt{\lambda}\|\boldsymbol{\theta}^*\|,$$

*where $\mathbf{U}_t = \lambda \mathbf{I}_d + \sum_{s=1}^t \mathbf{y}_s \mathbf{y}_s^\mathsf{T}$, $\boldsymbol{\nu}_t = \sum_{s=1}^t r_s \mathbf{y}_s$, $\boldsymbol{\theta}_t = \mathbf{U}_t^{-1} \boldsymbol{\nu}_t$, and*

$$\beta_t = 16\rho\sqrt{v_t \log(4t^2/\delta)} + 6\rho R \log(4t^2/\delta),$$

*where $\rho \geq \sup_{t \geq 1} \|\mathbf{y}_t\|_{\mathbf{U}_{t-1}^{-1}}$.*

## B.2   Proof of supporting lemmata

Lemmas B.2 and B.3 show that the concentration events $\mathcal{E}_{t,\ell}^{\boldsymbol{\theta}}$ and $\mathcal{E}_{t,\ell}^{\mathrm{TS}}$ hold with high probability for all $t \in [T]$ and $\ell \in [L]$.

**Lemma B.2.** *For any $\delta \in (0,1)$, if we define $\widetilde{\beta}_{t,\ell}$ for all $t \geq 1$ and $\ell \in [L]$ as in (B.1), then the estimate $\widehat{\boldsymbol{\theta}}_{t,\ell}$ given by the associated weighted regressions in Algorithm 3 satisfy*

$$\mathbb{P}\left( |\mathbf{x}^\mathsf{T}\widehat{\boldsymbol{\theta}}_{t,\ell} - \mathbf{x}^\mathsf{T}\boldsymbol{\theta}^*| \leq \widetilde{\beta}_{t,\ell}\|\mathbf{x}\|_{\mathbf{V}_{t,\ell}^{-1}}, \forall \mathbf{x} \in \mathcal{X}_t, \forall t \in \Psi_{t+1,\ell}, \forall \ell \in [L] \right) \geq 1 - \frac{\delta}{2}.$$

*Proof.* Notice that for any fixed $\ell \in [L]$, $t \in \Psi_{t+1,\ell}$, and $\mathbf{x} \in \mathcal{X}_t$ that

$$|\mathbf{x}^\mathsf{T}\widehat{\boldsymbol{\theta}}_{t,\ell} - \mathbf{x}^\mathsf{T}\boldsymbol{\theta}^*| \leq \|\widehat{\boldsymbol{\theta}}_{t,\ell} - \boldsymbol{\theta}^*\|_{\mathbf{V}_{t,\ell}}\|\mathbf{x}\|_{\mathbf{V}_{t,\ell}^{-1}}.$$

Following the assumption on $\varepsilon_t$, it holds that

$$|w_t\varepsilon_t| \leq |\varepsilon_t| \leq R, \quad \mathbb{E}[w_t\varepsilon_t \mid \mathscr{H}_t] = 0, \quad \mathbb{E}[w_t^2\varepsilon_t^2 \mid \mathscr{H}_t] \leq w_t^2\sigma_t^2, \quad \|w_t\mathbf{x}_t\|_{\mathbf{V}_{t,\ell}^{-1}} = 2^{-\ell},$$

where $w_t = 2^{-\ell}/\|\mathbf{x}_t\|_{\mathbf{V}_{t,\ell}^{-1}} \leq 1$. Apply Theorem B.1 with stochastic process $\{w_t\mathbf{x}_t, w_t\varepsilon_t\}$ to get that with probability at least $1 - \frac{\delta}{2L}$ for all $t \in \Psi_{T+1,\ell}$

$$\|\widehat{\boldsymbol{\theta}}_{t,\ell} - \boldsymbol{\theta}^*\|_{\mathbf{V}_{t,\ell}} \leq \widetilde{\beta}_{t,\ell},$$

where

$$\widetilde{\beta}_{t,\ell} = 16 \cdot 2^{-\ell}\sqrt{\sum_{s \in \Psi_{t,\ell}} w_s^2\sigma_s^2 \log(8t^2L/\delta)} + 6 \cdot 2^{-\ell}R\log(8t^2L/\delta) + 2^{-\ell+1}.$$

It follows from a union bound that with probability at least $1 - \delta/2$ that for all $\ell \in [L]$, $t \in \Psi_{t+1,\ell}$, and $\mathbf{x} \in \mathcal{X}_t$

$$|\mathbf{x}^\mathsf{T}\widehat{\boldsymbol{\theta}}_{t,\ell} - \mathbf{x}^\mathsf{T}\boldsymbol{\theta}^*| \leq \widetilde{\beta}_{t,\ell}\|\mathbf{x}\|_{\mathbf{V}_{t,\ell}^{-1}}.$$

$\square$

**Lemma B.3.** *For any $\delta \in (0,1)$ and $\widehat{\beta}_{t,\ell}$ defined in (5.3) for all $t \geq 1$ and $\ell \in [L]$, the sampled variables $\boldsymbol{\theta}_{t,\ell}^{\mathrm{TS}}$ in Algorithm 3 satisfy*

$$\mathbb{P}\left( |\mathbf{x}^\mathsf{T}\boldsymbol{\theta}_{t,\ell}^{\mathrm{TS}} - \mathbf{x}^\mathsf{T}\widehat{\boldsymbol{\theta}}_{t,\ell}| \leq \sqrt{2d\log(4t^2L/\delta)} \cdot \widehat{\beta}_{t,\ell}\|\mathbf{x}\|_{\mathbf{V}_{t,\ell}^{-1}}, \forall \mathbf{x} \in \mathcal{X}_t, \forall t \in \Psi_{t+1,\ell}, \forall \ell \in [L] \right) \geq 1 - \frac{\delta}{2}.$$

*Proof.* Note that $\boldsymbol{\theta}_{t,\ell}^{\mathrm{TS}} \sim \mathcal{N}(\widehat{\boldsymbol{\theta}}_{t,\ell}, \boldsymbol{\Sigma}_{t,\ell})$ where $\boldsymbol{\Sigma}_t = \widehat{\beta}_{t,\ell}^2 \mathbf{V}_{t,\ell}^{-1}$, we have with probability at least $1 - \frac{\delta}{4t^2L}$

$$\begin{aligned}
|\mathbf{x}^\mathsf{T}\boldsymbol{\theta}_{t,\ell}^{\mathrm{TS}} - \mathbf{x}^\mathsf{T}\widehat{\boldsymbol{\theta}}_{t,\ell}| &= |\mathbf{x}^\mathsf{T}\mathbf{V}_{t,\ell}^{-1/2}\mathbf{V}_{t,\ell}^{1/2}(\boldsymbol{\theta}_{t,\ell}^{\mathrm{TS}} - \widehat{\boldsymbol{\theta}}_{t,\ell})| \\
&\leq \widehat{\beta}_{t,\ell}\left\| \frac{\mathbf{V}_{t,\ell}^{1/2}(\boldsymbol{\theta}_{t,\ell}^{\mathrm{TS}} - \widehat{\boldsymbol{\theta}}_{t,\ell})}{\widehat{\beta}_{t,\ell}} \right\|_2 \|\mathbf{x}\|_{\mathbf{V}_{t,\ell}^{-1}} \\
&\leq \widehat{\beta}_{t,\ell}\sqrt{2d\log(4t^2L/\delta)}\|\mathbf{x}\|_{\mathbf{V}_{t,\ell}^{-1}},
\end{aligned}$$

where the last inequality follows the concentration inequality in Lemma C.1 for standard normal distribution and that

$$\left\| \frac{\mathbf{V}_{t,\ell}^{1/2}(\boldsymbol{\theta}_{t,\ell}^{\mathrm{TS}} - \widehat{\boldsymbol{\theta}}_{t,\ell})}{\widehat{\beta}_{t,\ell}} \right\|_2 \sim \mathcal{N}(0,1).$$

A union bound over all $\ell \in [L]$ and $t \geq 1$ yields the desired result, where the probability bound follows $\sum_{t=1}^{\infty} 1/t^2 < 2$. $\qquad\square$

Next, we show that the empirical estimator $\zeta_{t,\ell}$ provides an accurate estimate of the summation of weighted variance up to a logarithmic factor. The proof is analogous to Zhao et al. (2023), and we provide it here for the sake of maintaining a self-contained discourse.

**Lemma B.4.** *For all $t \geq 1$ and $\ell \in [L]$ such that $2^\ell \geq 64\sqrt{\log(4(t+1)^2 L/\delta)}$, if $\mathcal{E}_{t,\ell}^{\boldsymbol{\theta}}$ is satisfied and $\zeta_{t,\ell} = \sum_{s \in \Psi_{t,\ell}} w_s^2(r_s - \langle \mathbf{x}_s, \widehat{\boldsymbol{\theta}}_{s,\ell} \rangle)^2$, then the following inequalities hold:*

$$\sum_{s \in \Psi_{t+1,\ell}} w_s^2 \sigma_s^2 \leq 8\zeta_{t+1,\ell} + 6R^2 \log(4(t+1)^2 L/\delta) + 2^{-2\ell+4},$$

$$\zeta_{t+1,\ell} \leq \frac{3}{2} \sum_{s \in \Psi_{t+1,\ell}} w_s^2 \sigma_s^2 + \frac{7}{3} R^2 \log(4t^2 L/\delta) + 2^{-2\ell}.$$

*Proof.* For any $\ell \in [L]$ and $t \in [T]$, we have $\sum_{s \in \Psi_{t+1,\ell}} w_i^2 \varepsilon_s^2$ being an unbiased estimator of $\sum_{s \in \Psi_{t+1,\ell}} w_i^2 \sigma_s^2$ conditioned on past observations. More specifically, we have $\mathbb{E}[\varepsilon_t^2 \mid \mathbf{x}_{1:t}, r_{1:t-1}] - \mathbb{E}[\sigma_t^2 \mid \mathbf{x}_{1:t}, r_{1:t-1}] = 0$ for all $t \geq 1$ and

$$\sum_{s \in \Psi_{t+1,\ell}} \mathbb{E}[w_s^2(\varepsilon_s^2 - \sigma_s^2)^2 \mid \mathbf{x}_{1:t}, r_{1:t-1}] \leq \sum_{s \in \Psi_{t+1,\ell}} \mathbb{E}[w_s^2 \varepsilon_s^4 \mid \mathbf{x}_{1:t}, r_{1:t-1}]$$

$$\leq R^2 \sum_{s \in \Psi_{t+1,\ell}} w_s^2 \sigma_s^2,$$

where the first inequality follows from $\mathrm{Var}(w_s \varepsilon_s^2 \mid \mathbf{x}_{1:t}, r_{1:t-1}) \leq \mathbb{E}[w_s^2 \varepsilon_s^4 \mid \mathbf{x}_{1:t}, r_{1:t-1}]$. Following Freedman inequality, it holds that with probability at least $1 - 2\delta/L$ that for all $t \geq 1$

$$\left| \sum_{s \in \Psi_{t+1,\ell}} w_s^2 \varepsilon_s^2 - \sum_{s \in \Psi_{t+1,\ell}} w_s^2 \sigma_s^2 \right| \leq \sqrt{2R^2 \sum_{s \in \Psi_{t+1,\ell}} w_s^2 \sigma_s^2 \log(4t^2 L/\delta)} + \frac{4}{3} R^2 \log(4t^2 L/\delta)$$

$$\leq \frac{1}{2} \sum_{s \in \Psi_{t+1,\ell}} w_s^2 \sigma_s^2 + \frac{7}{3} R^2 \log(4t^2 L/\delta), \tag{B.2}$$

where the second inequality follows from Young's inequality.

It then follows that

$$\sum_{s \in \Psi_{t+1,\ell}} w_s^2 \sigma_s^2 \leq 2 \sum_{s \in \Psi_{t+1,\ell}} w_s^2 \varepsilon_s^2 + \frac{14}{3} R^2 \log(4t^2 L/\delta)$$

$$\leq 4 \sum_{s \in \Psi_{t,\ell}} w_s^2(r_s - \langle \mathbf{x}_s, \widehat{\boldsymbol{\theta}}_{t,\ell} \rangle)^2 + \frac{14}{3} R^2 \log(4t^2 L/\delta)$$

$$+ 4 \sum_{s \in \Psi_{t,\ell}} w_s^2((r_s - \langle \mathbf{x}_s, \widehat{\boldsymbol{\theta}}_{t,\ell} \rangle) - \varepsilon_s)^2,$$

where the second inequality follows from $(x+y)^2 \leq 2x^2 + 2y^2$. Notice that the last term is bounded by

$$\sum_{s \in \Psi_{t,\ell}} w_s^2((r_s - \langle \mathbf{x}_s, \widehat{\boldsymbol{\theta}}_{t,\ell} \rangle) - \varepsilon_s)^2 = \sum_{s \in \Psi_{t,\ell}} w_s^2 \langle \mathbf{x}_s, \widehat{\boldsymbol{\theta}}_{t,\ell} - \boldsymbol{\theta}^* \rangle^2$$

$$\leq (\widehat{\boldsymbol{\theta}}_{t,\ell} - \boldsymbol{\theta}^*)^\mathsf{T} \mathbf{V}_{t+1,\ell}(\widehat{\boldsymbol{\theta}}_{t,\ell} - \boldsymbol{\theta}^*)$$

$$\leq \widetilde{\beta}_{t+1,\ell}^2,$$

where the first inequality follows from $w_s^2 x_s x_s^{\mathsf{T}} \preceq \mathbf{V}_{t+1,\ell}$ and the second inequality is due to Lemma B.2. If we have $2^\ell \geq 64\sqrt{\log(8t^2L/\delta)}$, then

$$\widetilde{\beta}_{t+1,\ell}^2 \leq \frac{1}{8}\sum_{s\in\Psi_{t+1,\ell}} w_s^2\sigma_s^2 + 2(6\cdot 2^{-\ell}R\log(4(t+1)^2L/\delta) + 2^{-\ell+1})^2$$

and

$$\sum_{s\in\Psi_{t+1,\ell}} w_s^2\sigma_s^2 \leq 4\sum_{s\in\Psi_{t,\ell}} w_s^2(r_s - \langle\mathbf{x}_s,\widehat{\boldsymbol{\theta}}_{t,\ell}\rangle)^2 + \frac{14}{3}R^2\log(4t^2L/\delta)$$

$$+ \frac{1}{2}\sum_{s\in\Psi_{t+1,\ell}} w_s^2\sigma_s^2 + 8(6\cdot 2^{-\ell}R\log(4(t+1)^2L/\delta) + 2^{-\ell+1})^2.$$

Move $\frac{1}{2}\sum_{s\in\Psi_{t+1,\ell}} w_s^2\sigma_s^2$ to the other side of the inequality, we get the desired result.

Moreover, we can establish the inequality in the other direction:

$$\sum_{s\in\Psi_{t,\ell}} w_s^2(r_s - \langle\mathbf{x}_s,\widehat{\boldsymbol{\theta}}_{t,\ell}\rangle)^2 \leq \sum_{s\in\Psi_{t,\ell}} w_s^2(r_s - \langle\mathbf{x}_s,\boldsymbol{\theta}^*\rangle)^2 + 2^{-2\ell}\|\boldsymbol{\theta}^*\|_2^2$$

$$\leq \sum_{s\in\Psi_{t+1,\ell}} w_s^2\varepsilon_s^2 + 2^{-2\ell},$$

where the first inequality follows from $\widehat{\boldsymbol{\theta}}_{t,\ell}$ being the minimizer of the weighted ridge regression problem. Combining the inequality with (B.2) yields the desired result. □

**Lemma B.5.** *Suppose that* $\mathcal{E}_{t,\ell}^{\boldsymbol{\theta}}$ *and* $\mathcal{E}_{t,\ell}^{\mathrm{TS}}$ *hold for all* $\ell \in [L]$ *and* $t \in \Psi_{t+1,\ell}$. *If all the elements in series* $\{\widetilde{\beta}_{t,\ell}\}_{t,\ell}$ *and* $\{\widehat{\beta}_{t,\ell}\}_{t,\ell}$, *as defined in* (B.1) *and* (5.3) *respectively, satisfy*

$$\widetilde{\beta}_{t,\ell} \leq \widehat{\beta}_{t,\ell},$$

*then for all* $t \geq 1$ *and* $\ell \in [L]$ *such that* $\mathcal{X}_{t,\ell}$ *exists, it holds* $\mathbf{x}_t^* \in \mathcal{X}_{t,\ell}$.

*Proof.* We prove the statement by induction. If $\ell = 1$, then $\mathbf{x}_t^* \in \mathcal{X}_{t,\ell} = \mathcal{X}_t$ trivially holds. Suppose $\mathbf{x}_t^* \in \mathcal{X}_{t,\ell}$ for some $\ell \in \mathbb{Z}_+$ and $\mathcal{X}_{t,\ell+1}$ exists, we have for all $\mathbf{x} \in \mathcal{X}_{t,\ell}$ that

$$|\langle\mathbf{x}, \boldsymbol{\theta}_{t,\ell}^{\mathrm{TS}} - \boldsymbol{\theta}^*\rangle| \leq |\langle\mathbf{x}, \widehat{\boldsymbol{\theta}}_{t,\ell} - \boldsymbol{\theta}^*\rangle| + |\langle\mathbf{x}, \boldsymbol{\theta}_{t,\ell}^{\mathrm{TS}} - \widehat{\boldsymbol{\theta}}_{t,\ell}\rangle|$$

$$\leq (\sqrt{2d\log(4t^2L/\delta)} + 1)\widehat{\beta}_{t,\ell}\|\mathbf{x}\|_{\mathbf{V}_{t,\ell}^{-1}}$$

$$\leq 2^{-\ell}(\sqrt{2d\log(4t^2L/\delta)} + 1)\widehat{\beta}_{t,\ell},$$

where the second inequality is due to the assumption $\widetilde{\beta}_{t,\ell} \leq \widehat{\beta}_{t,\ell}$, Lemmas B.2 and B.3, and the last inequality is due to $\|\mathbf{x}\|_{\mathbf{V}_{t,\ell}^{-1}} \leq 2^{-\ell}$ for all $\mathbf{x} \in \mathcal{X}_{t,\ell}$. If we define $\widetilde{\mathbf{x}}_t = \arg\max_{\mathbf{x}\in\mathcal{X}_{t,\ell}}\langle\mathbf{x}, \boldsymbol{\theta}_{t,\ell}^{\mathrm{TS}}\rangle$, the it holds that

$$\langle\widetilde{\mathbf{x}}_t - \mathbf{x}_t^*, \boldsymbol{\theta}_{t,\ell}^{\mathrm{TS}}\rangle \leq \langle\widetilde{\mathbf{x}}_t - \mathbf{x}_t^*, \boldsymbol{\theta}^*\rangle + \langle\widetilde{\mathbf{x}}_t - \mathbf{x}_t^*, \boldsymbol{\theta}_{t,\ell}^{\mathrm{TS}} - \boldsymbol{\theta}^*\rangle$$

$$\leq |\langle\widetilde{\mathbf{x}}_t, \boldsymbol{\theta}_{t,\ell}^{\mathrm{TS}} - \boldsymbol{\theta}^*\rangle| + |\langle\mathbf{x}_t^*, \boldsymbol{\theta}_{t,\ell}^{\mathrm{TS}} - \boldsymbol{\theta}^*\rangle|$$

$$\leq 2\cdot 2^{-\ell}(\sqrt{2d\log(4t^2L/\delta)} + 1)\widehat{\beta}_{t,\ell},$$

where the second inequality follows from $\langle\widetilde{\mathbf{x}}_t - \mathbf{x}_t^*, \boldsymbol{\theta}^*\rangle \leq 0$. Hence, the optimal action $\mathbf{x}_t^* \in \mathcal{X}_{t,\ell+1}$ and the statement follows from induction. □

**Lemma B.6.** *Suppose that* $\widetilde{\beta}_{t,\ell} \leq \widehat{\beta}_{t,\ell}$, $\mathcal{E}_{t,\ell}^{\boldsymbol{\theta}}$, *and* $\mathcal{E}_{t,\ell}^{\mathrm{TS}}$ *hold for all* $\ell \in [L]$ *and* $t \in \Psi_{t+1,\ell}$. *For any* $\ell \in [L]\backslash\{1\}$, *we have*

$$\sum_{t\in\Psi_{T+1,\ell}} \Delta_t \leq 16d\cdot 2^\ell(\sqrt{2d\log(4T^2L/\delta)} + 1)\widehat{\beta}_{T,\ell-1}\cdot\log(1 + 2^{2\ell}T/d).$$

*Proof.* For all $t \in \Psi_{T+1,\ell}$, we have $\mathbf{x}_t, \mathbf{x}_t^* \in \mathcal{X}_{t,\ell}$ following Lemma B.5 and the assumptions on $\widetilde{\beta}_{t,\ell} \le \widehat{\beta}_{t,\ell}$, $\mathcal{E}_{t,\ell}^{\boldsymbol{\theta}}$, and $\mathcal{E}_{t,\ell}^{\text{TS}}$. Due to Line 11 in Algorithm 3, it holds

$$\langle \mathbf{x}_t^*, \boldsymbol{\theta}_{t,\ell-1}^{\text{TS}} \rangle - \langle \mathbf{x}_t, \boldsymbol{\theta}_{t,\ell-1}^{\text{TS}} \rangle \le 2 \cdot 2^{-\ell+1} (\sqrt{2d\log(4(t^2 L/\delta))} + 1) \widehat{\beta}_{t,\ell-1}. \tag{B.3}$$

In addition, for any $t \in \Psi_{T+1,\ell}$, we have both $\|\mathbf{x}_t\|_{\mathbf{V}_{t,\ell-1}^{-1}} \le 2^{-\ell+1}$ and $\|\mathbf{x}_t^*\|_{\mathbf{V}_{t,\ell-1}^{-1}} \le 2^{-\ell+1}$. It follows that

$$\begin{aligned}
\Delta_t &\le \langle \mathbf{x}_t^*, \boldsymbol{\theta}_{t,\ell-1}^{\text{TS}} \rangle - \langle \mathbf{x}_t, \boldsymbol{\theta}_{t,\ell-1}^{\text{TS}} \rangle + |\langle \mathbf{x}_t^*, \boldsymbol{\theta}_{t,\ell-1}^{\text{TS}} - \boldsymbol{\theta}^* \rangle| + |\langle \mathbf{x}_t, \boldsymbol{\theta}_{t,\ell-1}^{\text{TS}} - \boldsymbol{\theta}^* \rangle| \\
&\le 2 \cdot 2^{-\ell+1} (\sqrt{2d\log(4t^2 L/\delta)} + 1) \widehat{\beta}_{t,\ell-1} + 2 \cdot 2^{-\ell+1} (\sqrt{2d\log(4t^2 L/\delta)} + 1) \widehat{\beta}_{t,\ell-1} \\
&\le 8 \cdot 2^{-\ell} (\sqrt{2d\log(4t^2 L/\delta)} + 1) \widehat{\beta}_{t,\ell-1},
\end{aligned}$$

where the second inequality is due to (B.3), Lemmas B.2 and B.3. Summing over $t \in \Psi_{T+1,\ell}$, we have

$$\begin{aligned}
\sum_{t \in \Psi_{T+1,\ell}} \Delta_t &\le 8 \cdot 2^{-\ell} (\sqrt{2d\log(4T^2 L/\delta)} + 1) \widehat{\beta}_{T,\ell-1} |\Psi_{T+1,\ell}| \\
&\le 8 \cdot 2^{\ell} (\sqrt{2d\log(4T^2 L/\delta)} + 1) \widehat{\beta}_{T,\ell-1} \cdot \sum_{t \in \Psi_{T+1,\ell}} \|w_t \mathbf{x}_t\|_{\mathbf{V}_{t,\ell}^{-1}}^2 \\
&\le 8 \cdot 2^{\ell} (\sqrt{2d\log(4T^2 L/\delta)} + 1) \widehat{\beta}_{T,\ell-1} \cdot 2d\log(1 + 2^{2\ell} T/d),
\end{aligned}$$

where the second inequality is due to $\|w_t \mathbf{x}_t\|_{\mathbf{V}_{t,\ell}^{-1}} = 2^{-\ell}$ for all $t \in \Psi_{T+1,\ell}$, and the last inequality is due to Lemma C.2. $\qquad \square$

## B.3 Proof of Theorem 5.1

With probability at least $1 - 2\delta$, all the statements of Lemmas B.2 and B.3 hold, and the rest of our proof will build on such an event. Let $\ell_0 = \lceil \frac{1}{2} \log_2 \log(8(T+1)^2 L/\delta) \rceil + 8$, and we have $\widetilde{\beta}_{t,\ell} \le \widehat{\beta}_{t,\ell}$ for all $t \in [T]$ and $\ell \in [L] \backslash [\ell_0]$ following Lemma B.4. It then follows Lemma B.6 that for all $\ell \in [L] \backslash [\ell_0]$

$$\begin{aligned}
\sum_{t \in \Psi_{T+1,\ell}} \Delta_t &\le \widetilde{\mathcal{O}}(d^{3/2} 2^{\ell} \widehat{\beta}_{T,\ell-1}) \\
&\le \widetilde{\mathcal{O}} \left( d^{3/2} \sqrt{\sum_{t=1}^{T} w_t^2 (r_t - \langle \mathbf{x}_t, \widehat{\boldsymbol{\theta}}_{t+1,\ell} \rangle)^2 + R^2 + 1} + R \right) \\
&\le \widetilde{\mathcal{O}} \left( d^{3/2} \sqrt{\sum_{t=1}^{T} \sigma_t^2} + d^{3/2} R + d^{3/2} \right),
\end{aligned}$$

where the first inequality is due to Lemma B.6, the second inequality is due to the definition of $\widehat{\beta}_{T,\ell-1}$, and the last inequality is due to Lemma B.4. For any layer $\ell \in [\ell_0]$ and $t \in \Psi_{T+1,\ell}$, we have

$$\begin{aligned}
\sum_{t \in \Psi_{T+1,\ell}} \Delta_t &\le 2|\Psi_{T+1,\ell}| \\
&= 2^{2\ell+1} \sum_{t \in \Psi_{T+1,\ell}} \|w_t \mathbf{x}_t\|_{\mathbf{V}_{t,\ell}^{-1}}^2 \\
&\le \widetilde{\mathcal{O}}(d),
\end{aligned}$$

where the first inequality follows from $\langle \mathbf{x}, \boldsymbol{\theta}^* \rangle \le 1$ for all $\mathbf{x} \in \mathcal{X}_t$ and $t \in [T]$, the equality is due to $\|w_t \mathbf{x}_t\|_{\mathbf{V}_{t,\ell}^{-1}} = 2^{-\ell}$ for all $t \in \Psi_{T+1,\ell}$, and the last inequality is due to Lemma C.2 and $2^{\ell_0} \lesssim \sqrt{\log(TL/\delta)}$.

For any $t \in [T] \backslash (\cup_{\ell \in [L]} \Psi_{T+1,\ell})$, Algorithm 3 selects an arm at some $\ell_t$-th layer, and it follows that

$$
\sum_{t \in [T] \backslash (\cup_{\ell \in [L]} \Psi_{T+1,\ell})} \Delta_t \leq \sum_{t \in [T] \backslash (\cup_{\ell \in [L]} \Psi_{T+1,\ell})} (\langle \mathbf{x}_t^*, \boldsymbol{\theta}_{t,\ell_t}^{\mathrm{TS}} \rangle - \langle \mathbf{x}_t, \boldsymbol{\theta}^* \rangle)
$$

$$
+ \sum_{t \in [T] \backslash (\cup_{\ell \in [L]} \Psi_{T+1,\ell})} \sqrt{2d \log(4t^2 L / \delta)} + 1) \widehat{\beta}_{t,\ell_t} \| \mathbf{x}_t^* \|_{\mathbf{V}_{t,\ell_t}^{-1}}
$$

$$
\leq \sum_{t \in [T] \backslash (\cup_{\ell \in [L]} \Psi_{T+1,\ell})} (\langle \mathbf{x}_t, \boldsymbol{\theta}_{t,\ell_t}^{\mathrm{TS}} - \boldsymbol{\theta}^* \rangle + \alpha(\sqrt{2d \log(4t^2 L / \delta)} + 1) \widehat{\beta}_{t,\ell_t})
$$

$$
\leq \sum_{t \in [T] \backslash (\cup_{\ell \in [L]} \Psi_{T+1,\ell})} (\alpha \| \boldsymbol{\theta}_{t,\ell_t}^{\mathrm{TS}} - \boldsymbol{\theta}^* \|_{\mathbf{V}_{t,\ell_t}} + \alpha(\sqrt{2d \log(4t^2 L / \delta)} + 1) \widehat{\beta}_{t,\ell_t})
$$

$$
\leq \sum_{t \in [T] \backslash (\cup_{\ell \in [L]} \Psi_{T+1,\ell})} 2\alpha(\sqrt{2d \log(4t^2 L / \delta)} + 1) \widehat{\beta}_{t,\ell_t}
$$

$$
\leq T \cdot \widetilde{\mathcal{O}}(\sqrt{d}/T)
$$

$$
\leq \widetilde{\mathcal{O}}(\sqrt{d}),
$$

where the first inequality is due to $\widetilde{\beta}_{t,\ell} \leq \widehat{\beta}_{t,\ell}$, Lemmas B.2, B.3 and B.5, the second inequality is due to the arm selection rule of Algorithm 3, the third inequality is due to Cauchy-Schwarz inequality, and the fourth inequality is due to $\widetilde{\beta}_{t,\ell} \leq \widehat{\beta}_{t,\ell}$, Lemmas B.2 and B.3, and the fifth inequality follows from the definition of $\alpha$ and $\widehat{\beta}_{t,\ell_t} \leq \widetilde{\mathcal{O}}(R\sqrt{T})$.

Combining the above inequalities together, we conclude that

$$
\mathcal{R}(T) = \sum_{t=1}^{T} \Delta_t
$$

$$
\leq \widetilde{\mathcal{O}} \left( d^{3/2} + d^{3/2} R + d^{3/2} \sqrt{\sum_{t=1}^{T} \sigma_t^2} \right),
$$

where the inequality follows from $L \lesssim \log T$ and $\ell_0 = \lceil \frac{1}{2} \log_2 \log(8(T+1)^2 L / \delta) \rceil + 8$.

## C Auxiliary Lemmata

**Lemma C.1** (Abramowitz et al. (1964)). *For a Gaussian distributed random variable $X$ with mean $m$ and variance $\sigma^2$, it holds that for any $x \geq 1$ that*

$$
\frac{1}{2\sqrt{\pi}x} \exp(-x^2/2) \leq \mathbb{P}(|X - m| > x\sigma) \leq \frac{1}{\sqrt{\pi}x} \exp(-x^2/2).
$$

**Lemma C.2** (Abbasi-Yadkori et al. (2011), Lemma 11). *Let $\{\mathbf{X}_t\}_{t=1}^{\infty}$ be a sequence in $\mathbb{R}^d$, $\mathbf{V} \in \mathbb{R}^{d \times d}$ positive definite matrix, and $\overline{\mathbf{V}}_t = \mathbf{V} + \sum_{s=1}^{t} \mathbf{X}_s \mathbf{X}_s^{\mathsf{T}}$. Then we have that*

$$
\log \left( \frac{\det(\overline{\mathbf{V}}_n)}{\det(\mathbf{V})} \right) \leq \sum_{t=1}^{n} \| \mathbf{X}_t \|_{\overline{\mathbf{V}}_{t-1}^{-1}}^2.
$$

*Further, if $\| \mathbf{X}_t \|_2 \leq L$ for all $t$, then*

$$
\sum_{t=1}^{n} \min \left\{ 1, \| \mathbf{X}_t \|_{\overline{\mathbf{V}}_{t-1}^{-1}}^2 \right\} \leq 2(\log \det(\overline{\mathbf{V}}_n) - \log \det(\mathbf{V}))
$$

$$
\leq 2(d \log((\mathrm{Tr}(\mathbf{V}) + nL^2)/d) - \log \det(\mathbf{V})),
$$

*and finally, if $\lambda_{\min}(\mathbf{V}) \geq \max(1, L^2)$ then*

$$
\sum_{t=1}^{n} \| \mathbf{X}_t \|_{\overline{\mathbf{V}}_{t-1}^{-1}}^2 \leq 2 \log \frac{\det(\overline{\mathbf{V}}_n)}{\det(\mathbf{V})}.
$$

## D Numerical Results

In our simulation, we test our algorithm on a linear contextual bandit problem with different sets of reward perturbations. Specifically, the simulation is carried out over $T = 2000$ rounds with an ambient dimension of $d = 25$. For each selection round $t \in [T]$, the environment yields a set of contexts $\mathcal{X}_t$, which contains $K = 50$ context vectors. The context vectors are randomly drawn and structured as truncated Gaussian vectors, ensuring their norms remain bounded by $1$. Within each trial, a random ground truth vector $\boldsymbol{\theta}^*$ is generated as another Gaussian vector. The reward associated with every context vector is then computed based on the expected reward, perturbed by Gaussian noise. We employ our proposed algorithms, `LinVDTS` and `LinNATS`, to scrutinize their performance within this configuration.

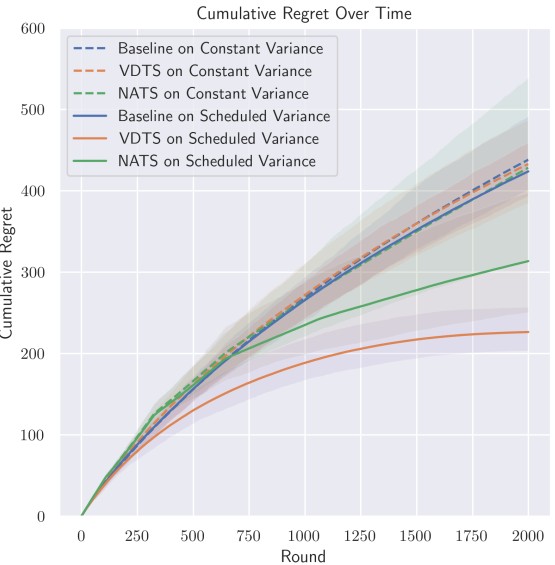

Figure 1: Performance comparison of `LinVDTS`, `LinNATS`, and the vanilla TS algorithms on the linear contextual bandit problem. The blue (`LinVDTS`), orange (`LinNATS`), and green (vanilla TS) lines depict the average regret for each method. Dashed lines represent the average regret in a constant variance scenario, while solid lines indicate the average regret under diminishing noise variance. Confidence bands, plotted for each method, are derived from 50 random trials.

Figure Fig. 1 presents the cumulative regrets of the `LinVDTS` and `LinNATS` algorithms, using the vanilla TS as a benchmark. The algorithms are evaluated in two distinct bandit environments: one characterized by constant variance and the other by a quadratically decaying variance. Performance under the constant variance setup is captured by dashed lines, whereas the decaying variance scenario is depicted by solid lines, with the observations being grounded on $50$ randomized trials. It is noteworthy that while `LinVDTS` is privy to the ground truth variance, `LinNATS` operates without this knowledge in our experimental framework. This distinction underscores `LinNATS`'s commendable adaptability in environments where the variance is elusive. Given its access to variance data, the superior performance of `LinVDTS` vis-a-vis `LinNATS` in fluctuating variance situations is anticipated. The presented outcomes showcase the variance-awareness of our proposed algorithms and the inherent flexibility of these algorithms in modulating their regret according to the prevailing variance dynamics.

**Future directions and broader impact.** The recent progress in understanding the training dynamics of deep neural networks (Jacot et al., 2018; Song et al., 2021) has paved the way for investigations at the intersection of deep learning and reinforcement learning (Lillicrap et al., 2015; Chen et al., 2021; Xu et al., 2021). This integrative approach promises to yield transformative insights and substantially enhance the performance of existing learning architectures. Further, the incorporation of non-linear function approximation into our frameworks emerges as a pivotal research direction,

with the potential to significantly improve their robustness (Ling et al., 2019; Chen et al., 2020; Min et al., 2021a; Ye et al., 2023). We believe that engaging in these research endeavors is important for the refinement of our methodologies.

