# OpenReview forum: "Noise-Adaptive Thompson Sampling for Linear Contextual Bandits"
_NeurIPS.cc/2023/Conference — NeurIPS 2023 poster_

### Official Review · Reviewer_xGkR · 2023-07-05

**Soundness:** 3 good
**Presentation:** 3 good
**Contribution:** 2 fair
**Rating:** 3
**Confidence:** 4

**Summary:**

This paper considers linear bandit problems with adaptive noise sequences. They propose LinVDTS that takes the noise level as input, applies a weighted RLS to estimate the parameter, and adopts a TS manner to choose the actions. LinVDTS achieves a noise-dependent regret that matches the best $O(d^{\frac{3}{2}} \sqrt{d})$ frequentist order. They also propose LinNATS, which drops the requirement of the knowledge of heteroscedastic noise using a weighted RLS estimator with multi-layer uncertainty structures. LinNATS achieves similar regret bounds.

**Strengths:**

The paper is well-organized and easy to follow. The proofs seem to be technically sound. In general, I think this paper does a great and complete job in providing theoretical guarantees for the noise-adaptive linear bandit problems.

**Weaknesses:**

My main concern is the significance and contributions of the job. The key contributions are claimed to be LinVDTS, LinNATS and their regret bounds. However, the RLS estimator and its proof technique are standard in the literature of Linear Thomson Sampling (LinTS). Compared with LinTS, the only new component of LinVDTS is the weighted RLS estimator (which is also standard in its literature) to handle adaptive noise. The remaining parts, including the associated regret proof, are essentially the same. The multi-layer design of LinNATS is also common in papers dealing with unknown parameters/uncertainty/non-stationarity. It's not a surprise the authors could get the presented results

I don't agree with the third contribution the author claimed: "This improves over the existing $\mathcal{O}(d^{\frac{3}{2}}\sqrt{T})$ regret bound for LinTS". When all variances are fixed at constant, the bound is still $\mathcal{O}(d^{\frac{3}{2}}\sqrt{T})$. The mentioned ``diminishing variances'' are too simplistic and shall not be over-emphasized.

Third, the presented Bayesian regret is of order $\mathcal{O}(d^{\frac{3}{2}}\sqrt{T}+\delta T)$, which is worse than the known best achievable $\mathcal{O}(d\sqrt{T})$ [1].

Refs:
1. Daniel Russo and Benjamin Van Roy. Learning to optimize via posterior sampling. Mathematics of Operations Research, 39(4):1221–1243, 2014. doi: 10.1287/moor.2014.0650. 2, 3

**Questions:**

As discussed above in the Weakness part, I would appreciate it if the authors could explain more regarding the significance and contributions of this work. Also, can the authors comment on why they fail to achieve optimal Bayesian bound?

**Limitations:**

I don't see any limitations or potential negative societal impact of this work.

---

> ### Author Rebuttal · Authors · 2023-08-07
>
> We are grateful to the reviewer for the feedback. In the subsequent response, we aim to address the concerns raised.
>
> **Q: Explain more regarding the significance and contributions of this work**
>
> We appreciate the reviewer's feedback. While we recognize and appreciate the existing literature pointed out by the reviewer, we stress that our work significantly diverges in terms of the primary technical challenges addressed. This results in a unique contribution to the body of research in this area. We elaborate on our contributions as follows:
>
> _1. Identifying a Unique Challenge of Uncertainty Control to TS-based Algorithms Absent in UCB-based Algorithms_
>
> While the direct application of weighted ridge regression has been employed in UCB-based algorithms to solve contextual bandits with heteroscedastic noise, the issue of securing a high probability bound using this widely accepted ridge regression approach had not been previously identified. As we've elucidated in our discussion of the VDTS algorithm, the strategy of weighted ridge regression estimation coupled with anti-concentration TS analysis fails to concurrently provide a variance-dependent and high-probability regret bound.
>
> We identified that the core challenge lies in the design of the canonical TS paradigm. The lack of exact optimism in arm selection necessitates the coarse anti-concentration argument, which falls short in capturing the dependence on the variances even when they are known to the algorithm.
>
> _2. Distinct Purpose of the Stratification Design in Our Algorithm_
>
> While the multi-layer design has been utilized in UCB-based algorithms, its application in our algorithm serves a largely different purpose. In UCB algorithms, the multi-layer design is employed to address the issue of the unknown variance when applying self-normalized concentration inequalities. In contrast, we introduced a stratification approach in our work primarily due to the need for explicit control of the uncertainty quantities related to the selected contexts, an issue that does not exist in UCB-based algorithms. Without proper uncertainty control, it is impossible to bound over the most basic regret decomposition.
>
> Our work uncovers this limitation inherent in the widely accepted TS technique and proposes an approach to overcome it. We have shown that this challenge can be effectively addressed by employing stratification over uncertainty levels, which furnishes explicit control over uncertainty quantities. Moreover, such an algorithmic design eradicates the need for the anti-concentration argument prevalent in TS-type algorithms, a complex technique ubiquitously used in almost all regret analyses of TS algorithms.
>
> To sum up, we put forth the first variance-dependent result in the TS setting. During this process, we identify unique challenges within existing TS algorithmic design and regret analysis techniques and develop effective strategies to overcome these challenges, especially those concerning uncertainty control.
>
> **Q: The mentioned “diminishing variances” are too simplistic and shall not be over-emphasized.**
>
> We appreciate the chance to clarify that we did not position the performance of our algorithm in the diminishing variance setting as a primary contribution of our work. The example concerning diminishing variance merely serves illustrative purposes.
>
> Our primary contribution lies in the development of noise-adaptive algorithms with explicit variance-dependent regret bounds. This represents an advancement over existing work, as our proposed algorithm offers variance-dependent regret bounds where previous methods fall short.
>
> Thank you for bringing this to our attention. We will ensure to clarify this point in our upcoming revision to avoid any potential misunderstandings.
>
> **Q: Bayesian regret and optimal Bayesian bound**
>
> We would like to clarify that our regret guarantees fall within the frequentist perspective, wherein the expectation is not measured with respect to the prior, as in the case of Bayesian regret. Hence, the Bayesian regret of order $\mathcal{O}(d\sqrt{T})$ as specified in Russo and Van Roy (2014) is not directly comparable to our frequentist regret bound $\mathcal{O}(d^{\frac{3}{2}}\sqrt{T}+\delta T)$, given that the expectations are calculated against different probability measures.
>
> Specifically, our high-probability upper bound aligns with the lower bound for canonical TS provided in Hamidi and Bayati (2020) under both the constant variance case and deterministic reward scenarios. Though it trails the minimax lower bound by an unavoidable factor of $\sqrt{d}$, which comes from the conservative nature of the canonical TS approach, this presents an interesting avenue for future research. Integrating components from more explorative algorithms such as Feel-Good TS (Zhang, 2021) could potentially help attain minimax optimality.

---

> > ### Author Response · Authors · 2023-08-21
> >
> > Dear Reviewer xGkR,
> >
> > We are genuinely thankful for your time and effort in reviewing our paper.
> >
> > As the author-reviewer discussion period is about to conclude, we respectfully request your review of our rebuttal. If you feel that our responses have addressed your concerns, we would be grateful if you might consider raising the score.
> >
> > Once again, we deeply value your review and are committed to incorporating all of the feedback into our manuscript.
> >
> > Best,
> >
> > Author of 2987

---

### Official Review · Reviewer_AK5v · 2023-07-05

**Soundness:** 3 good
**Presentation:** 4 excellent
**Contribution:** 3 good
**Rating:** 7
**Confidence:** 4

**Summary:**

This work delves into the field of the linear contextual bandit, a widely used model for addressing sequential decision-making problems with broad applications in real-world scenarios. In this framework, the player selects an arm from a predefined set and receives a random reward from an unknown distribution upon pulling it. The objective of this learning problem is to minimize the cumulative regret, which is the sum of the differences between the rewards obtained and the rewards of the optimal arm (which is unknown to the player) across all rounds. In particular, Thompson sampling (TS) emerges as a prominent family of algorithms for solving the linear contextual bandit problem, with an extensive body of literature dedicated to analyzing their theoretical properties. TS has garnered considerable attention in research, making it an active area of exploration. This work specifically focuses on the linear Thompson Sampling (TS) algorithms, where existing studies have established an upper bound of $O(d^{3/2} \sqrt{T})$ for the regret in Linear TS algorithms. However, this bound lacks noise adaptivity, meaning it does not consider the variance in the reward distribution. Intuitively, the bandit learning problem becomes easier when the reward distribution has lower variance, and vice versa.

This work addresses the limitations of previous methods by providing the first noise adaptive linear TS algorithms and theoretically proves that it achieves a variance-aware regret upper bound. Specifically, two different algorithms are presented. The first algorithm, VDTS, is shown to achieve a variance-aware regret bound in expectation. This work briefly explains the technical difficulty in establishing a high-probability variance-aware bound by identifying two factors: (1) the optimism in VDTS is too conservative; (2) a purely technical challenge for controlling a difference term in the regret analysis. The work also presents another high-probability bound for VDTS, but it is not variance-aware.

This study then introduces a new algorithm, NATS, which successfully achieves a variance-aware regret bound in high probability. This is the first TS-based algorithm for linear contextual bandit that is noise adaptive. The key design of NATS is a layering technique, which is motivated by some UCB-based algorithms. On a high level, this technique divides the uncertainty quantities into different layers depending on their magnitudes. By doing so, a variance-dependent confidence radius for model parameters can be built, which eventually leads to a variance-aware regret bound. This differs from the anti-concentration argument adopted by existing work in analyzing TS-based algorithms.

**Strengths:**

1. This work provides a solid theoretical analysis of two proposed TS-based linear bandit algorithms with well-grounded motivation. In particular, the study on how to achieve a high-probability noise-adaptive regret bound is comprehensive. The following are my detailed comments.
2. The focus on noise-awareness is well-motivated. I think it is particularly interesting to have a result for TS that bridges the gap between deterministic reward and constant-variance reward. Furthermore, since such types of results have been developed for UCB-based algorithms, it is a natural question to ask whether this is achievable in a TS setting.
3. This work is the first work to establish a high probability regret bound with noise-adaptivity for linear TS-based algorithms. Although such kind of results has been achieved by UCB-based algorithms in quite a few existing works, there is a scarcity of results concerning TS-based algorithms. Since TS is believed to have higher computational efficiency than UCB, variance-awareness will make TS more favorable, especially in scenarios characterized by noise heteroscedasticity.
4. Naively adopting the algorithmic design based on weighted linear regression from UCB will not grant a high probability of regret bound in the TS setting, which is due to the technical barrier in the analysis as mentioned on page 6. Therefore, the work takes a different route by borrowing the idea of stratification over the uncertainty quantity, as introduced in Section 5.1, which results in a variance-dependent concentration inequality (eq 5.2).
5. As a theoretical work, this current manuscript is well-written. Most parts are relatively easy to follow. The paragraphs are eloquently composed, giving a clear explanation of the complicated algorithmic design of VDTS and NATS. For example, after main theorems 4.1 and 5.1, remarks and discussions are added to provide implications of the results. The discussion on page 6 regarding why VDTS cannot achieve a high-probability variance-aware bound is especially appreciated.

**Weaknesses:**

1. For theoretical papers on bandit algorithms, numerical simulations are usually quite necessary for demonstrating the empirical performance of the proposed algorithms. For this work, this is especially the case as its main theoretical contribution is a variance-aware regret bound which often involves subtlety. I would not doubt that NATS/VDTS will produce a sublinear ($O(\sqrt{T})$) cumulative regret. The key implication of this work is whether the algorithm can truly adapt to the noise level, but it is not immediately clear. For more specific questions, please refer to the Question section below.
2. This is more of a question than a weakness, but the work mentioned that the current regret bound’s dependence on dimension d is $O(d^{3/2}$), in comparison with $O(d)$ in the Feel-good TS algorithm. This makes me very curious about whether the current method can be improved to reach optimality in d. For example, if it is not possible, what technical barrier prevents the method from doing so? It would be helpful to have a bit more discussion on this point.

**Questions:**

1. I would be curious to see how the algorithm performs. The following questions are worth investigating:

​	 (a) How does the regret of NATS respond to the change in the noise level? For example, if one runs a numerical simulation on two synthetic bandit instances with different $\sigma^2$, will the regret curves manifest obvious differences?

(b) It seems that the absence of a high-probability variance-aware bound for VDTS is more of a technical issue. So, I am interested to know whether VDTS indeed has a worse performance than NATS.

2. This manuscript uses both terminologies “variance aware” and “noise adaptive” without an explicit definition (though it's not a big issue). In my understanding, they refer to the same property. So, I am just a bit curious why using two terms instead of one. Or perhaps an explicit definition/explanation can be added to prevent any confusion for future readers.

**Limitations:**

See Weaknesses.

---

> ### Author Rebuttal · Authors · 2023-08-07
>
> We extend our heartfelt thanks to the reviewer for their generous praise and acknowledgment of our theoretical contributions.
>
> **Q: Missing experiments**
>
> For insights regarding the results of our numerical experiments, kindly refer to our collective response provided to all reviewers.
>
> **Q: Improving regret bound’s dependence on dimension $d$**
>
> Indeed, as outlined in the final section of our paper, exploring algorithms that yield an improved regret bound with an $O(d)$ dependence on the ambient dimension presents an intriguing direction for future research. We are actively engaged in developing such algorithms. For instance, the algorithms based on Feel-Good TS (Zhang, 2021) necessitate a specific regularization of the squared loss. This process deviates significantly from our current approach in proving techniques. For canonical TS, our NATS algorithm enjoys a regret bound that aligns with the near-optimal lower bound for both a fixed variance scenario and a constant reward scenario.
>
> **Q: Terminologies “variance aware” and “noise adaptive”**
>
> This is an insightful observation. We intentionally use both the terms "variance aware" and "noise adaptive" in our paper, each serving a specific purpose. The term "noise adaptive" is used to characterize our algorithms, highlighting that they do not require knowledge of the true variance to achieve variance-dependent regret, thereby illustrating adaptivity to unknown noise. On the other hand, we utilize "variance aware" and "variance dependent" to depict the resulting concentration bounds and regret performance of the algorithms, as these metrics are explicitly dictated by the true variance. In our subsequent revision, we will unify the terminology by merging "variance aware" and "variance dependent".

---

> ### Comment · Reviewer_AK5v · 2023-08-19
> **Response to the Authors' Rebuttal**
>
> I appreciate the authors' efforts in conducting numerical experiments for addressing my concerns.  After reading all reviewers' comments and the author's response, I agree with the other two reviewers with positive comments. Overall, this paper is novel and technically solid.  Hence, I would like to maintain my original score and recommend accepting this paper.

---

> > ### Author Response · Authors · 2023-08-21
> >
> > Thank you for your feedback and the time invested in reviewing our paper. We are pleased our revisions addressed your concerns and grateful for your positive acknowledgment of our paper's strengths and your support for its acceptance.
> >
> > Warm regards, Authors

---

### Official Review · Reviewer_oXdZ · 2023-07-05

**Soundness:** 3 good
**Presentation:** 4 excellent
**Contribution:** 3 good
**Rating:** 7
**Confidence:** 4

**Summary:**

This paper investigates the contextual linear bandit model, a widely utilized framework for online decision-making problems. Specifically, it focuses on the heteroscedastic noise setting, where each reward is associated with a different variance. The authors propose a novel noise-adaptive Thompson sampling algorithm that establishes a regret upper bound dependent on the variance. This theoretical contribution not only encompasses the findings from the constant-variance and deterministic-reward regimes but also provides an interpolation between these two settings.

The paper presents two algorithms in total. The first algorithm, VDTS, achieves a variance-dependent regret bound in expectation. However, the authors acknowledge the inherent challenge of controlling uncertainty induced by posterior sampling, preventing them from obtaining a high-probability regret upper bound.

To address this limitation, the authors introduce the NATS algorithm, which employs stratification to partition uncertainty terms into distinct layers. By doing so, it bypasses the conventional anti-concentration argument typically employed in Thompson sampling analysis. The proposed NATS algorithm is proven to achieve a high-probability regret upper bound, surpassing existing bounds for linear Thompson sampling.

**Strengths:**

**Novelty**

This paper makes a significant contribution by introducing the first variance-aware Thompson sampling algorithm. Previous theoretical work solely focused on regret bounds that were independent of noise variance. The authors employ novel techniques, drawing inspiration from the stratification framework initially proposed for UCB-based algorithms.

**Theoretical Contribution**

The theoretical contributions of this paper are commendable. The introduction of a variance-aware bound represents a substantial improvement over existing bounds for linear Thompson sampling. Traditional bounds may prove suboptimal when the true noise variance is relatively small. Additionally, the authors' analysis incorporates stratification, enabling them to circumvent the commonly adopted anti-concentration argument found in classical Thompson sampling analyses.

**Clarity**

This paper is presented in a clear and coherent manner. The mathematical definitions are precise and well-defined. The VDTS and NATS algorithms are thoroughly explained and accompanied by demonstrations. The main theorems (Theorem 4.1 and 5.1) are followed by discussions regarding their implications.

**Weaknesses:**

This paper has no empirical results at all. For such kinds of theoretical bandit algorithms, numerical experiments are usually required to demonstrate the correctness of the proposed methods and the validity of the proved theorems.

**Questions:**

1. Theoretically, NATS is shown to achieve a high probability regret upper bound, while VDTS only in-expectation regret upper bound. I am interested in whether NATS and VDTS will perform differently (or similarly) in empirical results. A numerical simulation would be interesting to check.
2. In this paper, variance-awareness of the proposed algorithm is only proven theoretically. But I doubt whether this can really be realized empirically. For example, if we apply NATS to two bandit instances with different noise levels, would the observed regret really be smaller if the variance is smaller?

**Limitations:**

Please refer to the above Weaknesses.

---

> ### Author Rebuttal · Authors · 2023-08-07
>
> We deeply appreciate the reviewer's generous words and acknowledgement of our theoretical contributions.
>
> **Q: Empirical results**
>
> For insights regarding the results of our numerical experiments, kindly refer to our collective response provided to all reviewers.

---

> > ### Comment · Reviewer_oXdZ · 2023-08-18
> >
> > I appreciate the authors for including additional numerical results, which further validate the adaptivity of the proposed algorithms to scheduled noise with varying variance. The rebuttal has well-addressed my questions.
> > After carefully read other reviewers' comments and the authors' rebuttal, while another reviewer has raised concerns about the paper's novelty regarding the RLS estimator and its proof technique, this paper's major contribution lies in its insight on Thompson Sampling, which appears to be novel for the literature. Therefore, I agree with other two reviewers' opinions and would appreciate seeing the paper accepted at NeurIPS. I intend to raise my score accordingly.

---

> > > ### Author Response · Authors · 2023-08-18
> > > **Response by Authors**
> > >
> > > Thank you for the re-evaluation of our paper and your decision to adjust the score! We are grateful for your recognition of the novel insights on Thompson Sampling that our work brings to the literature, and we deeply appreciate your time, constructive criticism, and for your endorsement regarding the paper's acceptance.
> > >
> > > Warm regards, Authors

---

### Official Review · Reviewer_Fr5q · 2023-07-07

**Soundness:** 4 excellent
**Presentation:** 4 excellent
**Contribution:** 3 good
**Rating:** 7
**Confidence:** 4

**Summary:**

This paper presents a Thompson-Sampling style algorithm for linear contextual bandits under a heteroscedastic noise setting. When the heteroskedastic noise is known, they present a simple algorithm, LinVDTS, that achieves a variance-dependent regret upper bound, by utilizing a weighted ridge regression estimator. When the noise is unknown, they present a more general algorithm, LinNATS, that adapts to the unknown noise by using the idea of optimism together with elimination to stratify the uncertainty level.


**Strengths:**

The paper is well-written. The author provides a clear structure of the two settings and intuitions for their algorithm design. They also provide a good transition between their two algorithms from an easier to a harder problem setting. Their theoretical bound is impressive in the sense that it is not obvious if one could hit the same regret bound \tilde{O}(d^{3/2}\sqrt{T}) as the constant variance case, but their result recovers this bound. Also, the discussion of lines 207-239 is very insightful.


**Weaknesses:**

The explanation of certain details of the algorithm could be improved. For example, the stratification process in the paper is a bit complicated. A picture to explain it would help.
Also, the author could add more annotations to the description of their algorithms so that it is easier to read. As another note, the paper is missing experiments, although the theoretical contribution is solid and it could be considered as a theory paper.

Minor Comments:
1. Line 137: the dependence of the noise on x_t should be more emphasized. At first glance, the fact that the noise at time t depends on the arm x_t was hidden.
2. Check the sentence structure/grammar. Eg in Line 207


**Questions:**

1. When the noise is known, can you provide a more elaborate explanation of why you choose \bar{\sigma}_s instead of \sigma_s? Intuitively, if the variance of the noise is known, I would tend to just normalize the variance so that the problem reduces to a homoscedastic noise setting and use classical algorithms to solve it, and I’d like to know more about why that’s not enough.
2. Can the authors comment on a matching lower bound that incorporates the correct dependence on the changing variance?
3. Is there an off-by-1 error in line 166? Namely, the definition of \theta_t uses x_t, whereas \theta_t in the algorithms seems to only depend on the history up to time t-1.
4. I think the algorithm is computationally inefficient for large or infinite arm sets, so it is more of a theoretical interest but the practical side of it is less impactful. Do the authors think the algorithm can be made practical?

---

> ### Author Rebuttal · Authors · 2023-08-07
>
> We extend our sincere gratitude to the reviewer for their positive remarks on our study. In response to the comments, we will enhance our paper by elucidating the stratification strategy further and adding more comprehensive annotations to the algorithm description.
>
> **Q: Missing experiments**
>
> For insights regarding the results of our numerical experiments, kindly refer to our collective response provided to all reviewers.
>
> **Q: Why choose $\bar\sigma_s$ instead of $\sigma_s$; why not enough just normalizing the variance?**
>
> Direct normalization of the variance, while seemingly straightforward, isn't an effective strategy in this context. The elliptical potential lemma, which we utilize to manage the uncertainty of selected arms, cannot adequately handle exponentially large context vectors when normalized by an exponentially small variance. Therefore, opting for direct normalization could negatively impact the regret bound. The application of $\bar\sigma_s$ allows us to strike a balance between managing noise variance and controlling the uncertainty of the context vector. This balance ultimately results in our final regret bound.
>
> **Q: Matching lower bound that incorporates the correct dependence on the changing variance**
>
> Thank you for the insightful question! There's currently no known lower bound, even in the context of the UCB case. We hypothesize that a potential lower bound could be in the order of $\Omega\Big(d\sqrt{\sum_{t=1}^T \sigma_t^2}\Big)$. This would hold true under the condition of homoscedastic noise, whereby the lower bound would reduce to $\Omega(d\sqrt{T})$. This is a minimax lower bound already achieved by the Feel-Good TS algorithm (Zhang, 2021). Eliminating the additional $d^{1/2}$ factor would constitute a significant undertaking in future research endeavors, and we appreciate you pointing us in this intriguing direction.
>
> **Q: Subscript off by one?**
>
> We appreciate your keen observation. We will ensure it's corrected in the forthcoming revision of the manuscript.
>
> **Q: Computationally inefficiency for large or infinite arm sets**
>
> In practical scenarios, we can leverage parallel computing to compute individual reward estimates across a discrete set of arms. In cases with an infinite set of arms, it is also possible to undertake the arm selection and decision set construction process efficiently if the set of all feasible context vectors is convex. Given a convex decision set $\mathcal{X}\_{t,\ell}$, the succeeding level decision set $\mathcal{X}\_{t,\ell+1}$ will also be a convex set through intersection of an additional hyperplane. The optimizer can be computed using convex optimization schemes. We appreciate the reviewer bringing this point to our attention. We plan to incorporate a discussion regarding this in the final revision of our paper.

---

> > ### Comment · Reviewer_Fr5q · 2023-08-15
> > **Response**
> >
> > Thank you for your comment and for addressing my concerns. I will leave my review where it is.

---

> > > ### Author Response · Authors · 2023-08-15
> > > **Response by Authors**
> > >
> > > We appreciate the reviewer's feedback and their favorable assessment of our paper. If there are any additional inquiries from the reviewer, we would be happy for further discussion. Kindly let us know at your earliest convenience, as the authors' engagement in the discourse is limited by a specific deadline.
> > >
> > > Best,
> > > Authors

---

### Author Rebuttal · Authors · 2023-08-07

We sincerely thank all reviewers for their valuable feedback. We will address all highlighted points in the upcoming revision. Detailed responses to specific questions are included below and will also be reflected in the revised manuscript.

**Numerical simulation results**

In response to the collective request of the reviewers, we have integrated numerical experiments into our response. A detailed figure can be found in the attached PDF file.

Our simulation focuses on a linear contextual bandit problem incorporating different reward perturbations. In this setup, we put our proposed algorithms, VDTS and NATS, to the test under a bandit configuration with the total number of $T = 1000$ bandit selection rounds and the ambient dimension $d = 25$. In every selection round $t\in [T]$, the environment generates a context set, $\mathcal{X}_t$, comprising a random assortment of $K=50$ context vectors. These vectors are configured as truncated Gaussian vectors such that their norms do not exceed $1$. Concurrently, a random ground truth $\theta^*$ is generated as a truncated Gaussian vector within each trial. The reward corresponding to each context vector is then determined by the expected reward, offset by Gaussian noise. The complete simulation setup and more simulation results will be included in the revision.

The figure showcases the cumulative regrets of the VDTS and NATS algorithms, with the Vanilla TS serving as a benchmark for comparison. Each algorithm is tested under two different bandit setups: one with a constant variance and another with a quadratically decaying variance. The dashed lines depict the performance under a constant variance scenario, while the solid lines correspond to the decaying variance scenario. These results are based on 50 random trials.

It's worth noting that in our experimental setup, the VDTS algorithm has access to the ground truth variance, while NATS does not. Therefore, the performance of NATS truly reflects its adaptability under conditions of unknown variance. The superior performance of VDTS in comparison to NATS under a varying variance setup is to be expected, as VDTS has explicit access to variance information.

The results validate two critical aspects:
(i) The empirical realization of variance awareness in our proposed algorithms, and
(ii) The adaptive nature of the algorithms' regret in response to different variance scheduling.

**Q: Whether NATS and VDTS will perform differently (or similarly) in empirical results?**

Both NATS and VDTS perform similarly in our experiments under a constant variance setup, and VDTS achieves smaller cumulative regret when the variance decreases over time.

**Q: Whether variance awareness of the proposed algorithm can be realized empirically?**

Yes. As evidenced in the figure, the performance of both NATS and VDTS progressively improves as the variance of the perturbation decreases, thereby illustrating the empirical realization of variance awareness in our proposed algorithms.

**Q: Does the regret of NATS respond to the change in the noise level?**

Indeed, it does. As observable from the results, the regret associated with NATS markedly decreases as the level of variance diminishes, demonstrating the algorithm's responsive nature to changes in noise levels.

**Q: Whether VDTS indeed has a worse performance than NATS?**

Our numerical findings suggest that under heteroscedastic noise, VDTS typically outperforms NATS. This observation aligns with our expectations, given that VDTS has the advantage of access to explicit variance information, which NATS does not have.

---

### Decision · Program_Chairs · 2023-09-21

**Decision:**

Accept (poster)

**Comment:**

The paper makes a clear contribution of being adaptive to the unknown noise for the Thompson sampling algorithm. The paper is well-written, providing insight.